# A single-cell atlas of the bobtail squid visual and nervous system highlights molecular principles of convergent evolution

Daria Gavriouchkina[1,2] ✉, Yongkai Tan[3], Elise Parey[4], Fabienne Ziadi-Künzli[5], Yuko Hasegawa[2,6], Laura Piovani[4], Lin Zhang[2], Chikatoshi Sugimoto[2,7], Nicholas Luscombe[3], Ferdinand Marlétaz[4] ✉ & Daniel S. Rokhsar[2,8,9,10] ✉

The cephalopod and vertebrate visual systems are a textbook example of convergent evolution with unknown molecular underpinnings. Here we characterize 98,537 single-cell transcriptomes in the bobtail squid *Euprymna berryi* to understand how the cephalopod retina and optic lobes relate to the vertebrate retina. We confirm the overall relative simplicity of the cephalopod retina but identify two related photoreceptor cell subtypes expressing distinct r-opsins. By contrast, the adult optic lobe contains a diverse repertoire of neuronal and glial cell types, with a predominance of dopaminergic neurons. We show that cephalopod-specific gene duplicates probably contributed to this cell type diversification. Comparing neuronal cell population in the optic lobes of hatchlings and adults, we reveal a switch towards dopaminergic neurotransmitter usage with age, indicative of a maturation process. We further identify an FMRF-amide-based retrograde signal from the optic lobe towards the retina that supports the functional analogy of the cephalopod optic lobe cortex and the vertebrate inner retina in visual signal processing from a molecular standpoint. Finally, comparative analyses with vertebrate and arthropod cells suggest a scenario in which two photoreceptor types and two neuronal populations may have already been present in the eye of the bilaterian ancestor.

Camera eyes of vertebrates and cephalopods are a classical example of convergent evolution, as an image-forming eye was absent in the last common ancestor of molluscs and chordates[1,2]. The visual systems of cephalopods and vertebrates, however, show marked differences in organization and function (Fig. 1a). In cephalopods, light reaches photoreceptors directly; by contrast, the vertebrate retina is 'inverted', so that light must traverse several cell layers before reaching photoreceptors. Cephalopods have a single type of rhabdomeric photoreceptor along with pigmented 'support' cells[3], while the vertebrate retina includes photoreceptors as well as additional neuronal cell types that process the visual signal before transmission to higher brain centres[4].

Cephalopod photoreceptor axons project directly to a pair of large bilateral optic lobes of the brain, which are organized into a layered outer cortex and inner medulla (Fig. 1a). A century ago, Santiago Ramón y Cajal made pioneering observations of neural cell morphology and organization in squid and octopus and noted similarities between the cortex of the cephalopod optic lobes and vertebrate inner retina, prompting him to call the cephalopod optic lobe cortex the 'deep retina' (*la retina profunda*). The hypothesis of an analogous functional role in visual signal processing to that of vertebrate inner retina[4] gained further support from the detailed ultrastructural studies of J. Z. Young in octopus and squid[5,6] further suggesting neuronal feedback from the optic lobe to photoreceptors[4–6].

Despite these differences, studies have shown that more than 60% of transcripts expressed in the octopus eye have orthologues expressed in vertebrate eyes[7]. These observations raised the question of whether convergent gene expression underlies structural and functional convergence. Using single-cell transcriptomics, we can now assess how homologous genes are deployed in various cell types of cephalopods and vertebrates and investigate potential cellular homologies[8]. Interestingly, there is still much debate over the complexity of the visual and nervous system in the last bilaterian ancestor, with many arguing in favour of a simple network of neurons[9]. Nevertheless, notable similarities in circuitry between visual systems of vertebrates and the fruit fly[10] have prompted theories regarding the cells and circuits in the ancestral bilaterian visual system[11–13]. The cephalopod and vertebrate visual systems, which we know a priori to be convergent, provide an ideal opportunity to begin to compare genes and cell types underlying convergently evolved complex systems.

Here, we apply single-cell transcriptomics and comparative genomics to investigate the cell type repertoire of the cephalopod retina and optic lobes and thereby evaluate the deep retina hypothesis (previously formulated from morphology) in terms of molecular signatures of cell identity. We propose the Japanese bobtail squid *Euprymna berryi* Sasaki 1929[14] (Fig. 1b) as an emerging cephalopod model[15] that is closely related to other bobtail squids[16,17] but is somewhat larger and more amenable to laboratory culture. We generated a high-quality reference genome and annotation for *E. berryi*. We confirm the simplicity of the cephalopod retina and observe a surprising diversity of optic lobe cell types. We provide evidence of a maturation process affecting optic lobe cell type complements that takes place after hatching. We evaluate the influence of gene duplication on the creation of novel cell types and convergent traits in cephalopods.

## Results

### A genomic resource for an amenable bobtail squid model

We assembled a highly contiguous chromosome-scale reference genome for *E. berryi* by combining long-read sequencing with chromatin conformation data (Extended Data Fig. 1a–c and Supplementary Table 1). Our assembly totals 5.9 Gb and captures 46 chromosomes found in decapods (N50 chromosome length 113.96 Mb, N50 contig length 827 kb)[18,19]. We recovered the massive 17 Mb Hox cluster as an intact locus on chromosome 9 (Extended Data Fig. 1f). Different expansions of long interspersed nuclear elements (LINE) and other repeats may explain the difference in genome size as compared with other cephalopod species[18–23] (Extended Data Fig. 1c). To support single-cell and comparative analysis, we generated a combination of bulk short-read and long-read RNA sequencing (RNA-seq; Supplementary Table 2). Using transcriptome data, we annotate 32,244 protein-coding genes in *E. berryi* with detectable homology in other species.

*E. berryi* chromosomes are generally derived from the fusion of several ancestral linkage groups (ALGs) as shown by comparison with the bivalve mollusc *Pecten maximus*: for instance, the tandem array-rich chromosome 20 corresponds to a mixture of ALGs I and G[19,24] (Fig. 1e,f). The 46 chromosomes of *E. berryi* show extensive conservation with the distantly related squid *Doryteuthis pealei* (Extended Data Fig. 1d)[19].

### A diverse neural gene complement in the bobtail squid

The *E. berryi* genome encodes an extensive complement of genes associated with neurotransmitter synthesis, degradation, transport and reception (Fig. 1f,g). While genes associated with neurotransmitter biosynthesis and degradation have remained in relatively stable copy numbers across bilaterians, we find extensive lineage-specific expansions in neurotransmitter receptors and transporters in *E. berryi*, but also in other spiralians (Fig. 1g). We see more variation in copy number for genes associated with acetylcholine and glutamate activity, whereas inhibitory GABA and glycine activity appears more uniform in copy number across bilaterians. For example, we identified single copies of monoamine-synthesizing enzymes such as tyrosine hydroxylase (*Ty3h*), tryptophan hydroxylase (*Tph2*) or tyrosine decarboxylase (*Tdc1*) across bilaterians. By contrast, the *E. berryi* genome encodes multiple ionotropic acetylcholine receptor subunits and multiple spiralian-specific families of glutamate receptors, both metabotropic (for example, *Grma-Grmi*) and ionotropic (for example, GluPhi, Phi-like and Delta)[25,26]. The absence of the glutaminase enzyme in *E. berryi* and other cephalopods suggests that glutamate synthesis is accomplished through another mechanism.

Expanded gene families are typically found in gene duplication hotspots, which are extensive across the genome (Fig. 1f): *E. berryi* shares an expansion of ionotropic acetylcholine receptors described in lophotrochozoans[26] located in two main hotspots on chromosomes 4 and 5. Interestingly, such parallel expansions occurred independently in other spiralian lineages (for example, annelids and molluscs; Fig. 1g and Extended Data Fig. 1g). We recover tandem arrays of C2H2 transcription factors (TFs) and protocadherins as reported in other cephalopod species[19] (Fig. 1g).

*E. berryi* therefore preserves the ancestral bilaterian complement of neural genes better than other lineages including subfamilies that have been lost in ecdysozoans or vertebrates (Fig. 1g). Some of these subfamilies are only present in spiralians (receptors: *AchS1-AchS3*, *mAchRC*, *Grmi1-2* and Grms, *Oar*, *Octa*, *Octb*, *Oamb*, *TyR*, *Htr6* and transporter *GlyT2*; Fig. 1g). However, it is noteworthy that other spiralians, such as annelids and lophophorates, have similarly diverse neural gene complements, which suggests that gene diversification alone is insufficient to explain cephalopod neural complexity.

### A single-cell atlas of the squid visual system

By applying single-cell (sc)RNA-seq, we characterized the transcriptomes of 98,537 cells, including the adult retina (15,223 cells), optic lobes (19,029 cells from 1-day-old hatchlings and 26,436 cells from mature adults) and non-visual organs (17,468 cells from perioesophageal

---

**Fig. 1 | Genomic organization and neural gene complement of a cephalopod. a**, A simplified representation of vertebrate and cephalopod camera-type eyes with ganglion cells (green), photoreceptor cells (yellow). **b**, An image of an adult *E. berryi*. **c**, Left: a micro-CT segmentation of the central nervous system of a hatchling *E. berryi*, in toto (1 dph). Right: a magnified view of the micro-CT segmentation of the central nervous system of an adult *E. berryi* (70 dph) — postero-dorsal view, indicating eyes, brain lobes and associated white body. **d**, A bar plot indicating the number of single cells sampled (right) and the number of biological replicates (left). The number of technical replicates used is indicated in Extended Data Fig. 2. OLs, optic lobes. **e**, An Oxford plot showing macrosyntenic blocks of orthologous genes located on *E. berryi* chromosomes (*y* axis) and a bivalve mollusc *Pecten maximus* (sea scallop) chromosomes (*x* axis.) Positions along each axis represent the integer-valued gene index. The colour code (to the left) corresponds to ALGs. Fusion events that took place in the cephalopod lineage can be observed at intersections of different ALGs.

**f**, The location of different categories of neural genes across the 46 chromosomes of *E. berryi*. **g**, A summary of phylogenetic analyses conducted for selected neural gene families and subfamilies stemming to the bilaterian ancestor involved in neurotransmitter synthesis, transport, degradation and reception. Monoamines include dopamine, serotonin, octopamine and tyramine. The colour code indicates activity (legend bottom right). ac, anterior chamber; co, cornea; cor, cortex; eso, oesophagus; ey, eye; ibl, inferior buccal lobe; ir, iris; le, lens; md, medulla; mu, muscle; ol, optic lobe; on, optic nerve; PSM, posterior suboesophageal mass; re, retina; sbl, superior buccal lobe; SBM, subesophageal mass; SPM, supraesophageal mass; vbl, posterior basal lobe; vtl, vertical lobe; wb, white body. Credits: photo in **b** courtesy of Ryuta Nakajima; silhouettes from PhyloPic under a Creative Commons license: human, brachiopod, sea slug and octopus, T. Michael Keesey; amphioxus, Guillaume Dera; *C. elegans*, Bob Goldstein and Jake Warner; fruit fly, Gareth Monger; annelid, Noah Schlottman (based on photo by Casey Dunn).

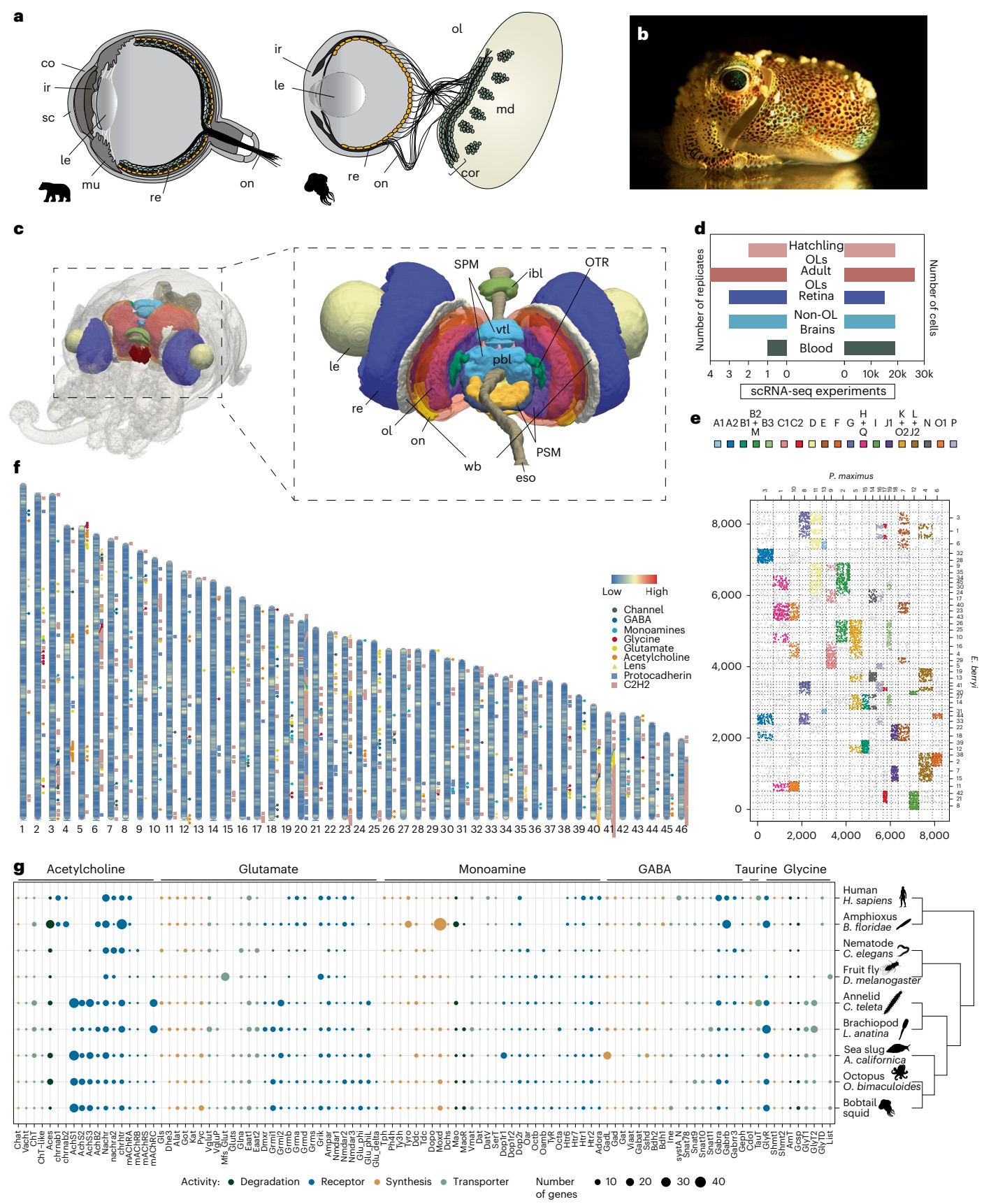

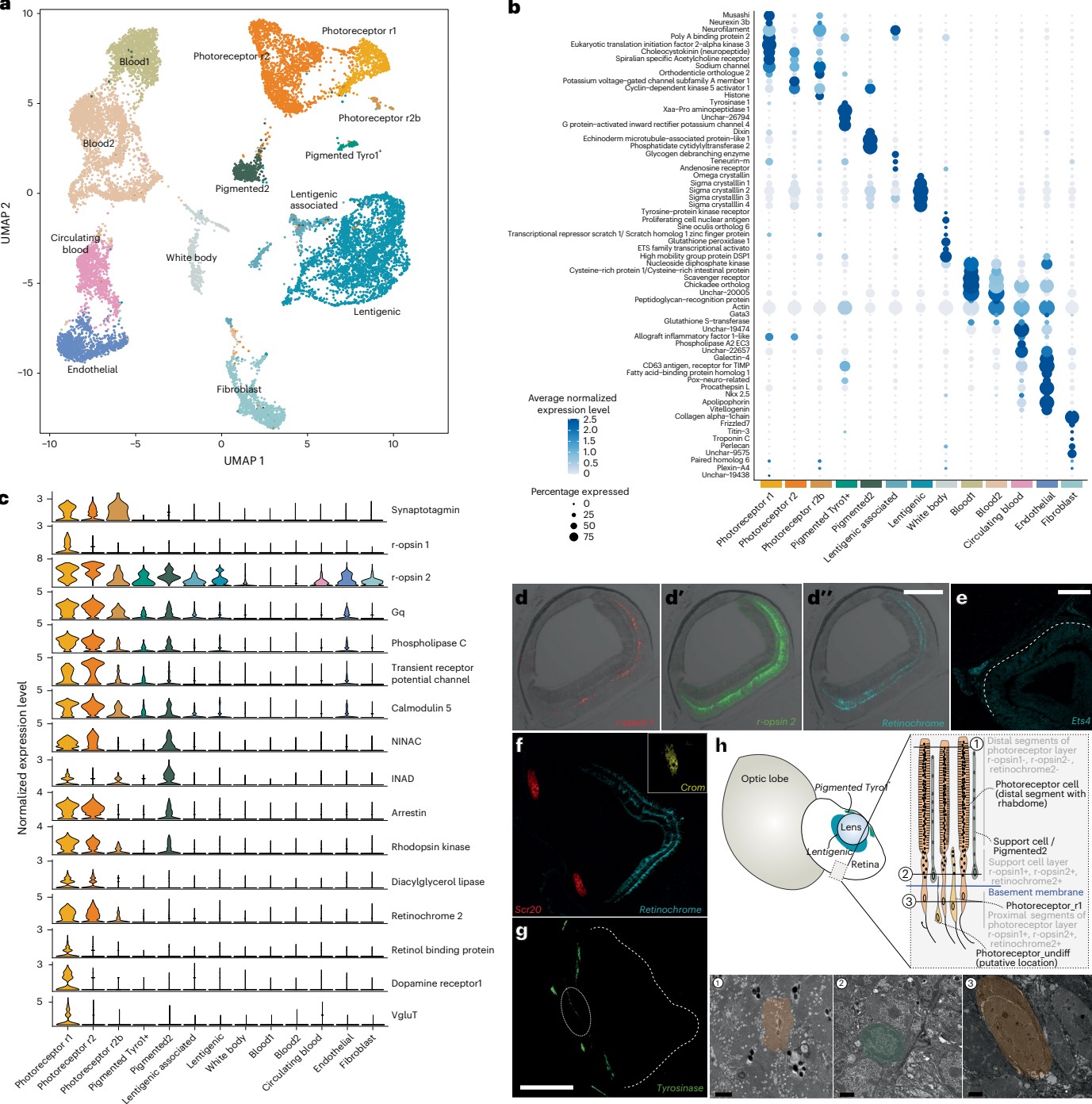

**Fig. 2 | Cell type complement of the *E. berryi* retina. a**, UMAP of 15,223 cells from dissected adult *E. berryi* retinas. **b**, A dot plot indicating genes specifically expressed in each set of cell clusters. **c**, Violin plots indicating expression levels of genes involved in phototransduction of rhabdomeric opsins and visual cycle. **d**–**g**, Fluorescence imaging of HCR stainings (*n* > 3 replications of stainings): three photoreceptive molecules r-opsin1 (**d**), r-opsin2 (**d′**) and retinochrome2 (**d″**), displaying expression demonstrating the localization of r-opsin1 in the support cell layer and r-opsin2 and retinochrome-2 in the support cell layer and in the proximal segment layer of the retina; Ets4 expression in the presumptive white body between the retina and optic lobe (**e**); retinochrome-2 and two lentigenic markers σ-crystallin (*Scr20-109*) and Ω-crystallin (*Crom*) (**f**); tyrosinase (*Tyro*) expression in tissues surrounding the pupil of the retina (**g**). **h**, A schematic of the retina indicating the putative location of cell clusters within retinal organization and images of a section through the retina at the level of the support cell layer, demonstrating the presence of multiple cell types at levels indicated in the schematic demonstrating (1) rhabdomes of photoreceptor cells, (2) support cells near the basement membrane and (3) proximal segments of photoreceptor cells featuring nuclei. Scale bars, 50 μm (**d**–**g**) and 2 μm (**h**).

adult brain and 20,381 blood cells from mature adults) (Fig. 1c,d and Extended Data Fig. 2). Neurons were identified as cells positive for neuronal markers synaptotagmin (*Sy65*), voltage-dependent calcium channel subunit alpha-1 (*Cac1a*) and a voltage-gated sodium channel (*Scna*) (Figs. 2b,c and 3b).

We found haemocytes in our retina, optic lobe and brain samples, which we identified by the expression the blood markers *Nkx25*, *Netr-1* and *Pgsc2* (Fig. 2b). The nature of these cells was confirmed by scRNA-seq conducted on whole blood from adult *E. berryi* (20,381 cells). A haematopoietic organ, the white body, is located between the retina

and optic lobe (Fig. 1c). Presumptive white body cells were identified by the co-expression of blood markers *Tie1* and *Pcna*[27,28].

## The bobtail squid retina displays few neuronal cell types

We captured 15,223 cells from dissected adult *E. berryi* retinas (Fig. 2a,b). We observe only four retinal cell types, of which two populations are photoreceptors (photoreceptors r1 and r2) distinguished by their differential expression of cephalopod-specific r-opsin paralogues (r-opsin1 and r-opsin2). In addition to opsins, both populations express phototransduction cascade effectors (*Gq*, *PLC* and arrestin) and retinol-recycling markers (Fig. 2c). Out of six putative *E. berryi* photosensitive GPCRs identified by phylogenetic analysis (rhabdomeric or r-opsins 1 and 2; retinochrome 1 and 2; xenopsin1 and 2)[29], only the two r-opsins and retinochrome-2 showed detectable retinal expression in both scRNA-seq and bulk RNA-seq (Extended Data Fig. 3b). r-opsin2 is expressed in non-visual tissues and cell types as in other cephalopods[30–32].

Photoreceptor r1 also expresses the retinol recycling RABP gene (Fig. 2c)[33–35], the stemness marker Musashi and several neuronal and neurotransmission genes including the vesicular glutamate transporter *Vglut* and dopamine receptor *Dopr1* (Fig. 2c). Expression of r-opsin1 is detected by hybridization chain reaction (HCR) below the basement membrane, suggesting that these cells may occupy a more proximal location in the retina (Fig. 2d–d″ and Supplementary Table 3). A small enigmatic Musashi+ population (photoreceptor r2b) also expresses *Sy65* and phototransduction markers (but not *RABP*), as well as an ephrin receptor *Epha7* previously described as a marker of developing photoreceptors in other cephalopods[36].

The rhabdomeric photoreceptors of *E. berryi* use glutamate as indicated by the expression of multiple glutamate-synthesizing enzymes (for example, Alat and Got) and transporters (for example, *Glut* and *VgluT*; Fig. 2c), similarly to vertebrate ciliary photoreceptors but differently from the histamine used in fly rhabdomeric photoreceptors[37]. We did not detect presynaptic markers for acetylcholine, serotonin, dopamine or octopamine, previously proposed to play a role in other cephalopods[38–41].

The 'Pigmented2' cell population is positive for phototransduction genes, but negative for the neuronal marker *Sy65* (Fig. 2c), and probably corresponds to the enigmatic support cells described in literature as pigmented and located underneath the retina basement membrane[36,42–46]. We observe a single support cell population in *E. berryi*, intermingled in the retina, which can be readily identified on corresponding electron microscopy sections (Fig. 2h). A second pigmented population ('Pigmented_tyro1') features tyrosinase (*Tyro*) in a ring-like domain around the aperture of the pupil (Fig. 2g).

Lentigenic cells located at the periphery of the retina secrete the lens composed of two classes of crystallins of independent phylogenetic origins: cephalopod-specific glutathione-*S*-transferase-derived lens proteins (σ-crystallins)[46–48] and spiralian-specific aldehyde dehydrogenase-like proteins (Ω-crystallins)[18,49]. We confirmed the

co-expression of both Ω-crystallin and σ-crystallins in lentigenic cells by HCR staining[47,50,51] (Fig. 2f). Two expansion hotspots on chromosomes 40 and 41 contribute 193 copies of σ-crystallins (Fig. 1f). Moreover, lentigenic cells also expressed *Sp9*, a Krüppel-like TF involved in *D. pealei* lentigenic cell specification[52].

## The optic lobe utilizes a complex cell type catalogue

The photoreceptor axons leave the retina in bundles surrounding the eye and enter the optic lobes latero-ventrally (Fig. 1a,c). Optic lobes are composed of a cortex (Cajal's *retina profunda*) and a medulla[3] (Fig. 3d). The cortex comprises two granule layers containing neuronal cell bodies and a plexiform layer containing neurites and no cell bodies, as shown by *Sy65* expression (Fig. 3e). Photoreceptor axons have been described to run through the outer granule layer without forming synapses and then synapse with elements of the plexiform layer or outer part of the medulla, the palisade layer[3,6] (Figs. 1a and 3d–l′).

We profiled 26,436 cells from mature adult optic lobes (>60 days post-hatching (dph); Extended Data Fig. 2a–d) and identified 26 major cell populations (Fig. 3a), of which 22 express pan-neuronal marker Elav4. Twenty Elav4+ clusters correspond to differentiated neuronal cell populations (Elav4+, Sy65+ and Scna+), two clusters (Neuro_1 and Neuro_2) constitute undifferentiated neuronal cells (Elva4+ but Sy65−) and four clusters represent non-neuronal types (Fig. 3a,b). The majority of optic lobe neuronal cell types are dopaminergic (12/22) expressing dopamine-synthesizing enzymes (*Ty3h* and *Ddc-2*) and transporters (*Dat-2* and *Vmat2-2*) (Fig. 3a,b). We identified two cholinergic clusters (expressing acetylcholine-synthesizing enzyme *ChAT*, acetylcholine transporter *VAchT*, choline transporter *Sc5a7-2* and acetylcholine-degrading enzyme *Aces-2*), three glutamatergic cell types (vesicular glutamate transporter VgluT+ and Vat1l+; Fig. 3a,b) and two putative GABAergic inhibitory clusters (GAT+, Gabt-1+ and Gabp2+; Fig. 3a) as well as a small monoaminergic cluster (Vmat2-2+). Each cluster expresses distinct complements of neurotransmitter receptors, implying that these neurons receive diverse inputs, including a cluster-specific set of acetylcholine and monoaminergic neurotransmitter receptors (Figs. 4f,g and 5i). It is curious that dopaminergic neurons also express low levels of *VgluT*, the primary vesicular glutamate transporter in *E. berryi*, reminiscent of observations of glutamate–dopamine co-transmission in vertebrates both during embryonic development and in adults as a toxin response[53].

## A retrograde signal involving dopamine and FMRF signalling

We used HCR to determine the putative localization of identified cell populations in optic lobes (Fig. 3c–l′; summary in Extended Data Fig. 4a). Distinct types of dopaminergic (that is, Ty3h+) neurons were found in both the cortex and the medulla (Fig. 3f). We assigned three dopaminergic populations (Dopaminergic_7, 10, 11; Fig. 3a,b) to the cortex outer granule layer based on the expression of TF *Six4* and kainate glutamate receptor *Grik2-4* (Fig. 3i,j). Other dopaminergic populations (Dopaminergic_5, 6 and Dopaminergic_1, 3, 9; see below)

---

**Fig. 3 | Cell type complement of the *E. berryi* adult optic lobe. a**, UMAP of 26,436 cells from an adult *E. berryi* optic lobe highlighting the location of each cell population. **b**, Violin plots indicating genes specifically expressed in sets of key cell clusters. **c**, Antibody staining demonstrating the location of FMRF-amide peptide, acetylated tubulin and 4′,6-diamidino-2-phenylindole (DAPI) in the *E. berryi* retina. **d**, Toluidine blue staining of a cross-section of the optic lobe. Inset: an overview of the optic lobe. **e–l′**, Fluorescence imaging of HCR stainings: pan-neuronal marker *Sy65* (synaptotagmin) expression demonstrating the location of neuronal cells in both the optic lobe cortex and the medulla (**e**); dopamine synthesis enzyme *Ty3h* (tyrosine hydroxylase) expression in a subset of neuronal cells of both the optic lobe cortex and the medulla (**f**); cholinergic cell marker, acetylcholine esterase, *Aces-2*, expression in cells of the optic lobe medulla (**g**); kainate glutamate receptor *Grik2-2* expression in the optic lobe outer granule layer (ogl) and in large cells of the medulla (**h**); kainate glutamate

receptor *Grik2-4* expression in the optic lobe outer granule layer and punctate expression in the medulla (**i**); *Six4* expression in the optic lobe outer granule layer of the cortex (**j**); FMRF ligand transcript localization using HCR in whole mounts (showing an overview of FMRF ligand producing cells arranged in a rosette formation located in deep cells of the medulla (**k**) and a magnification of FMRF-expressing cell clusters in the medulla (**k′**)); the expression of glial marker *Eaa1-2* on cryosection (**l**) and in whole-mount staining (**l′**), demonstrating *Eaa1-2* expression in large cells of the medulla including large cells located in the palisade layer of the medulla (located directly beneath the inner granule layer of cortex) and a punctate pattern in cells that may be adjacent to projections of cells in the inner plexiform layer and of axonal projections in the medulla. Stainings were reproduced at least three times. igl, inner granule layer; ipl, inner plexiform layer; md, medulla; ogl, outer granule layer. Scale bars, 100 μm (**c**) and 50 μm (**d–l′**).

are located in the medulla (*Grik2-2*[+], Fig. 3h). Cholinergic (*Aces-2*[+]) cells were observed in various locations within the medulla, including the palisade layer situated directly beneath the cortex (Fig. 3a,b,g). Notably, phylogenetically closely related genes exhibit distinct expression domains. Cephalopod-specific paralogues kainate glutamate receptors

*Grik2-2* and *Grik2-4* were observed in the outer granule layer and in large cells of the medulla with complementary expression patterns and in distinct cell clusters (*Grik2-2* in clusters Dopaminergic_1, 3, 9 and *Grik2-4* in the dopaminergic cells of the cortex (Dopaminergic_7, 10, 11) and centrifugal cells (Dopaminergic_5, 6 clusters, see below; Fig. 3a,b,k,k').

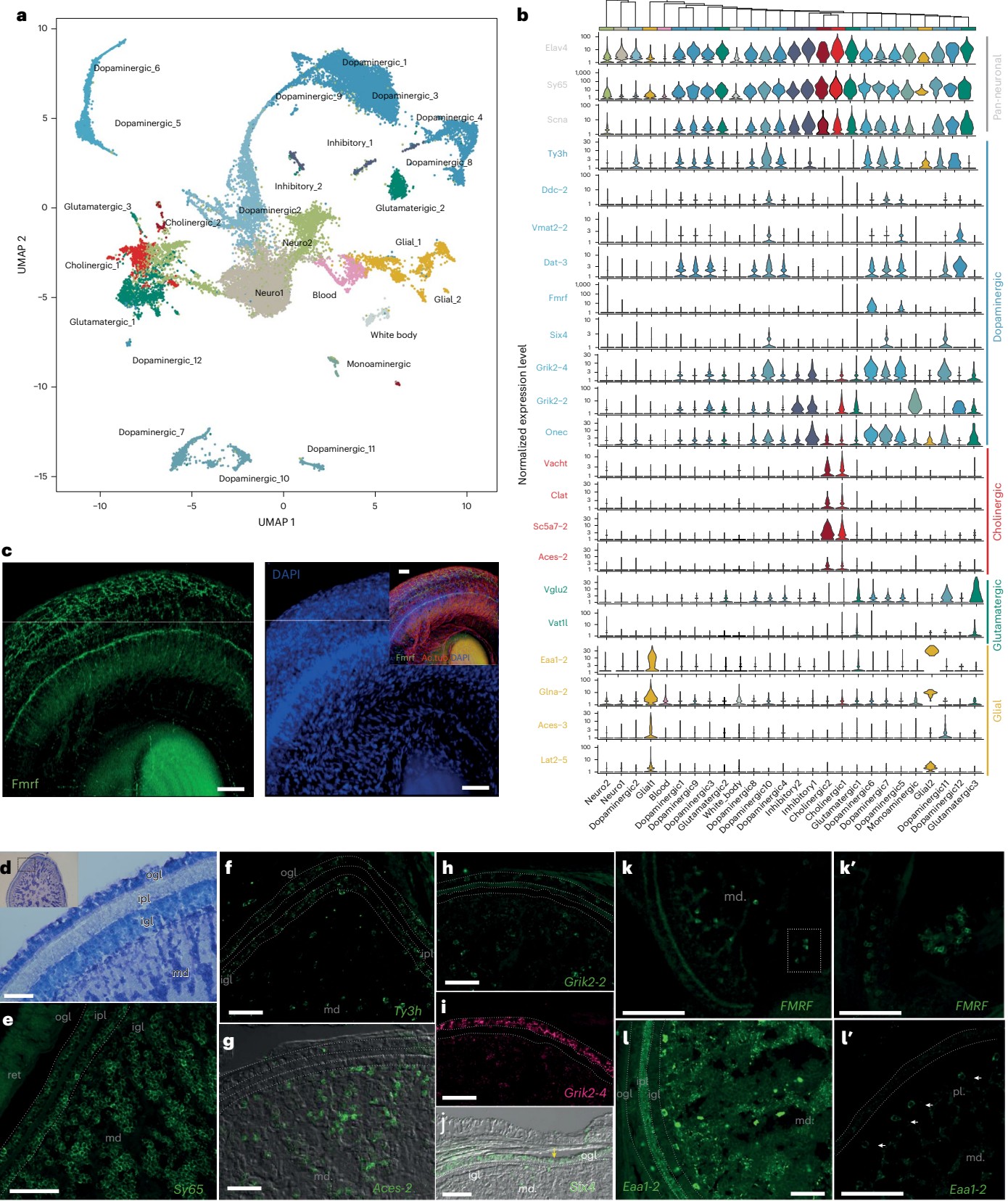

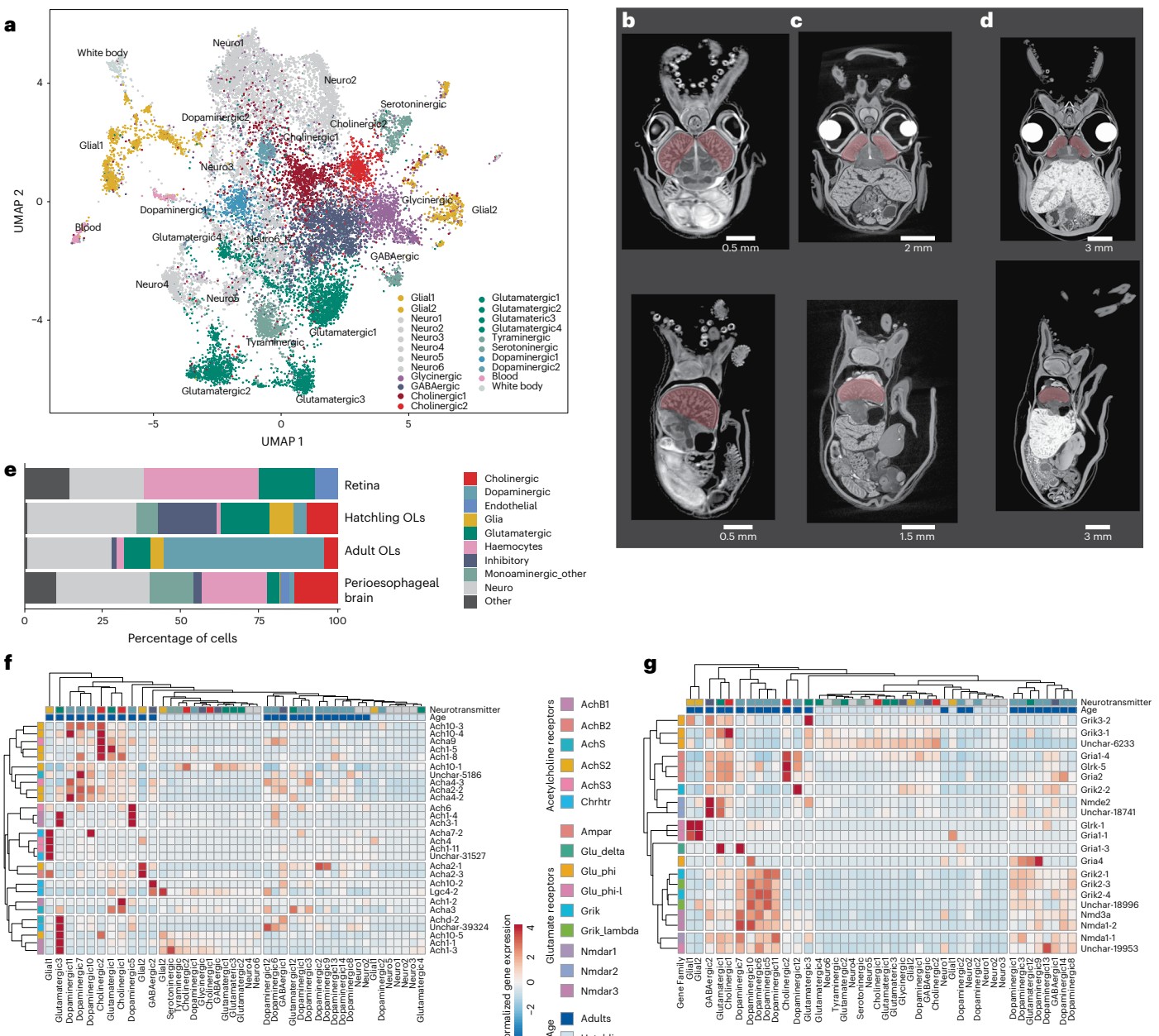

**Fig. 4 | Extended single-cell stage and organ profiling and optic lobe maturation. a**, UMAP of 19,029 cells from a 1-dph hatchling. **b–d**, A comparison of the optic lobe morphology by micro-CT segmentation and rendering of optic lobes 1 day after hatching (**b**), 40 days after hatching (**c**) and 80 days after hatching (**d**). **e**, A stacked bar plot of the number of cells of each cell type in each single-cell dataset. The inset shows an anterior view of the optic lobes at 1 dph with the minute optic nerve located ventrally (yellow) **f,g**, Heat maps showing expression levels of cholinergic (**f**) and glutamatergic (**g**) markers in adult and hatchling optic lobes.

Among the predictions of Cajal's deep retina hypothesis was that some cells in the cephalopod optic lobe medulla would project back to the retina[5,6]. We identified FMRF-amide as a potential retrograde signal from optic lobes to the retina in *E. berryi*. FMRF peptide was detected in the retina through antibody staining (Fig. 3c), but its encoding transcript was not found in the retina through scRNA-seq, bulk RNA-seq or HCR. We did, however, recover prominent expression of both transcript and peptide in the optic lobes (Fig. 3a,c). Dopaminergic populations 5 and 6 (top left of the uniform manifold approximation and projection (UMAP), Fig. 3a) produce FMRF-amide transcript and were observed using HCR in the cells of the optic lobe medulla, organized in a rosette formation (Fig. 3c,k,k'). These cells match the features of the centrifugal cells described by J. Z. Young[5], which is consistent with the suggestion of retrograde projections from the medulla to the retina[4–6].

This finding is also corroborated with recent whole-cell patch-clamp recordings from centrifugal neurons of the optic lobe showing reactivity to FMRF-amide[54]. These may be functionally analogous to the lateral inhibition of photoreceptors mediated by horizontal cells in the vertebrate retina relying on distinct signalling mechanisms[55].

## Identification of glial cell population

Glial cells are non-neuronal support cells that are present in most complex nervous systems but whose evolutionary origin and homology remain unclear[56]. We identified several putative glial cell types in *E. berryi* among non-neuronal (that is, *Elav4⁻*, *Sy65⁻* and *Scna⁻*) cell populations. Although there are no known glial markers in cephalopods[57,58], the glial identity of these cells is supported by the absence of any presynaptic markers and presence of neurotransmitter degrading enzymes

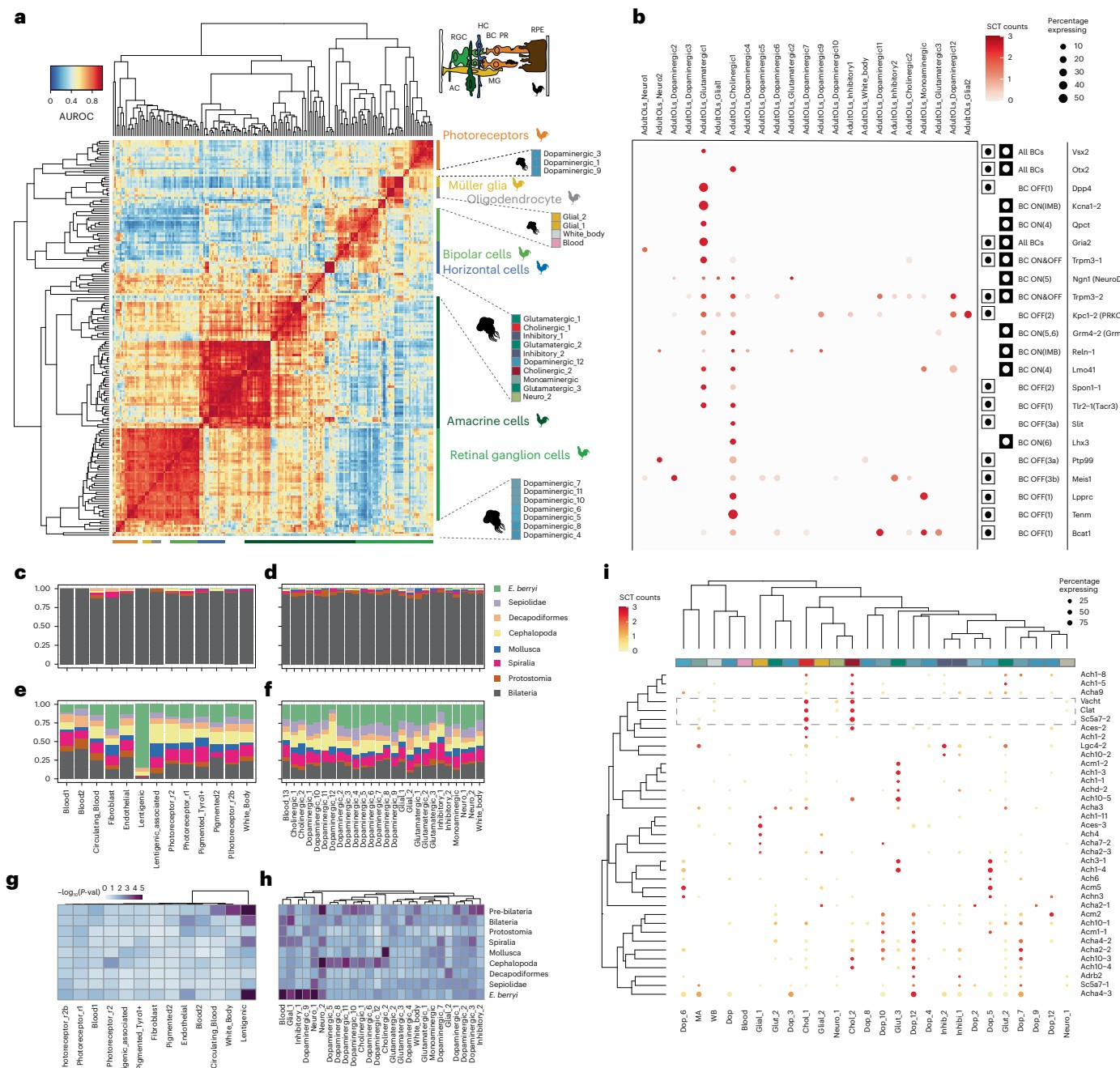

**Fig. 5 | Comparison of the cephalopod visual system gene expression with the vertebrate framework. a**, A heat map displaying AUROC scores or 'Reciprocal_top_hit' match types identified as similar cell-type pairs in comparisons between *E. berryi* adult optic lobe cell populations and cell types of the chicken retina[65]. Inset: the location of cell types in the vertebrate retina. **b**, A dot plot of the expression of selected vertebrate retinal bipolar cell marker orthologues in the adult *E. berryi* optic lobe. **c,d**, Stacked bar plots indicating the fraction of genes originating from gene families at each strata in retinal (**c**) and adult optic lobe (**d**) cell populations. **e,f**, Stacked bar plots indicating the fraction of genes whose last gene duplication dates to each strata in retinal (**e**) and adult optic lobe (**f**) cell populations. **g,h**, Heat maps displaying the significance of enrichment for a given phylostrata as −log$_{10}$(*P* value) of the Fisher's exact test (two-sided) for the enrichment of genes whose last duplication took place in each strata in retina (**g**) and adult optic lobe (**h**) cell populations. **i**, Dot plots indicating the expression levels of cholinergic neurotransmitter genes in the mature adult optic lobes of *E. berryi*. A box with dotted lines indicates the expression of presynaptic genes involved in the synthesis of neurotransmitters. RGC, retina ganglion cells; BC, bipolar cells; HC, horizontal cells; PR, photoreceptor cells; RPE, retina pigmented epithelium; MG, Müller glia; AC, amacrine cells. Credits: inset in **a** reproduced with permission from ref. [128], Elsevier; chicken silhouette by Steven Traver reproduced from PhyloPic under a Creative Commons license CC0 1.0.

(*Glna-2* and *Aces-3*) and transporters (*Lat2-5* and an *Eaat*), which have, in the case of *Eaat* transporters, been associated with both glial and presynaptic cells in vertebrates[59]. Our analysis provides a catalogue of marker genes enriched in glial cells of *E. berryi* (Supplementary Tables 4–7). Notably, we did not observe any expression of orthologues of arthropod glial markers such as *Repo* and *Gcm*, or evidence of the presence of vertebrate markers such as *GFAP* or *S100* in the *E. berryi* genome or in other cephalopod genomes[18,20], which supports the convergent evolution of glial cells across diverse bilaterians[56] (Supplementary Note 3).

The glial marker *Eaat1-2* is expressed in both large cells of the palisade layer of the medulla (Fig. 3j,j', arrowheads) and in a punctate pattern in the inner plexiform layer, also associated with putative neurites of the medulla reminiscent of glial cells in *Octopus*[60]. Interestingly, *Eaat* mRNAs have recently been localized to endosomes in vertebrates[61]. These two distinct patterns may correspond to the two distinct putative glial clusters in *E. berryi* optic lobes ('Glial_1' and 'Glial_2') (Fig. 3a,b,l,l'). Based on previous histological studies[60], we propose that the larger cells may correspond to fibrous glia, whereas the punctate pattern may correspond to protoplasmic glia (Supplementary Note 3).

## Cell type diversity in hatchlings suggests optic lobe maturation

The eyes and optic lobes of cephalopods grow considerably over the life of the animal[62] and underlie a range of age-specific visual behaviours. One-day-old *E. berryi* hatchlings exhibit light-sensitive behaviours such as burying and visually driven hunting of prey, implying that their visual system is functional (Supplementary Videos 1 and 2)[63]. Mature animals exhibit additional visual behaviours, including those relating to mating[64]. To evaluate changes in the visual system with age, we characterized the morphological differences in optic lobes using micro-computed tomography (micro-CT) and found differences in shape (from half-sphere to kidney-bean shape; Fig. 4b–d and Extended Data Fig. 5b,c) throughout their post-hatching life, but also a change in the position and size of the optic nerve and commissural neurons. Notably, the cortical thickness increases between 1 dph and 80 dph while the ratio of cortical and medullar volumes remained constant, indicating isometric growth of these two closely integrated structures (Fig. 1a,c, Extended Data Fig. 5 and Supplementary Table 8). To study the impact of these changes on cell types, we profiled 19,029 cells from the optic lobes of 1-day-old *E. berryi* hatchlings (Fig. 4a) and compared them with adult samples. Glial, haemocyte, white body and several neuronal populations express similar genes to those observed in adult optic lobes (Extended Data Fig. 5f–h). However, neurotransmitter usage differed substantially between hatchling and adult: the number of dopaminergic cells and cell types increased with age. We recovered fewer dopaminergic cells in hatchlings than in the mature optic lobe, with only two dopaminergic clusters, along with four glutamatergic, two cholinergic and six undetermined neuronal populations (Figs. 3a and 4a and Extended Data Fig. 5f–h). The combination of neurotransmitter receptors observed in mature optic lobes is different from the one in the hatchling optic lobe (Fig. 4f,g). We also observe tyraminergic and serotonergic cell types in hatchling optic lobes that were not captured in discernible numbers from adult optic lobes, as well as glycinergic inhibitory neurons.

To test whether increased usage of dopamine is a general hallmark of central nervous system maturation or a particularity of the optic lobes, we analysed an additional 17,468 cells from the peri-oesophageal brain lobes (that is, non-optic lobe brain) from the same mature adult individuals. We found a complex set of neuronal cell types (Extended Data Fig. 5a). Unlike the optic lobe, however, dopaminergic cells are not predominant, but there is an abundance of cholinergic cells. Although this dataset is insufficient to fully characterize the complexity of the cephalopod brain, it extends our classification of cell types based on neurotransmitter type beyond the optic lobe.

## The evolutionary origin of cephalopod visual system

To detect correspondence between visual system cell types of cephalopod and those of other species, we compared the cell types of the bobtail squid visual system with those of the chicken retina[65] (Fig. 5a, Extended Data Fig. 6a and Supplementary Table 9) and the adult fruit fly optic lobe[66] (Extended Data Fig. 6f) as well as those of mouse brain regions involved in visual processing, such as the dorsal lateral geniculate cortex[67] and visual cortex[68] (Extended Data Fig. 6b,c). Cell types were compared using transcriptome-wide gene expression similarity

of homologous genes through a pairwise unsupervised analysis (MetaNeighbor; Fig. 5a and Extended Data Fig. 6a–d)[69,70] and cross-species integration of single-cell profiles (SAMap; Extended Data Figs. 6e and 7a–d)[71]. Neuronal cell types are primarily grouped together by species and according to their neurotransmitter in the similarity-based clustering, and few groupings gather cell types from both species. Some excitatory neuron types, however, appear similar across species. In particular, dopaminergic cell types 1 and 3 of the bobtail squid optic lobe consistently display gene expression similarity with vertebrates resembling ciliary photoreceptors of the vertebrate retina (chicken; Fig. 5a and Extended Data Fig. 6e) and glutamatergic cells of the visual cortex and dorsal lateral geniculate nucleus (mouse; Extended Data Fig. 6b,c). Interestingly, vertebrate retinal ganglion cells show some affinity for the squid photoreceptor, which can be explained by their shared usage of rhabdomeric melanopsin and associated transduction pathways[72]. Moreover, inhibitory cell types appear less similar to each other than excitatory ones (for example, optic lobe glutamatergic cells). High similarity scores in cell types that are 'outgroups' to the neurons (glial cells, blood, white body and endothelium) may be caused by the weak similarity between neuronal and non-neuronal cells rather than by evolutionary conservation. Altogether, our cross-comparisons argue for a reduced repertoire of ancestral neuronal cell types in bilaterians, as they support the lineage-specific elaboration of each neurotransmitter-specific cell population.

We further investigated TFs proposed to be determinants of cell type identity conserved across lineages[73]. We found that haemocytes and endothelial cells express conserved mesodermal factors including *Klf5*, *Ets4*, *Nkx25*, *Etv6*, *Sox9* and *Gata3-1* (Extended Data Fig. 8c). Despite previous reports[74], *Ets4* appears to be a marker of the white body in our datasets and is not expressed in the optic lobe of *E. berryi*. By contrast, we did not observe a strong conservation of TFs that classically define eye-related cell types in other lineages. Canonical retinal determination genes[75,76] (*Atonal*, *Six*, *Eya* and *Pax6*) have limited co-expression in *E. berryi*: Pax6 is only subtly expressed in photoreceptors and in fibroblasts surrounding the eye as well as in the putative centrifugal-like cells in the optic lobe (Dopaminergic 5 and 6, also *Egl13*+; Extended Data Fig. 4b,c); sine oculis (*Six1/4/6*) is observed in the cortex of the optic lobe (together with *Grik2-2*) in cells that may correspond to the amacrine cells described by J. Z. Young[5,6] (Fig. 3j).

Two models have been previously proposed to explain cell type evolution at the origins of complex visual systems[11]. In a first model[73], all vertebrate retinal cells are proposed to have arisen from two spatially and molecularly distinct ancestral photoreceptors related to ciliary photoreceptors (bipolar cells) and rhabdomeric photoreceptors (retinal ganglion cells, horizontal cells and amacrine cells). In a second model[76], the ancestral eye already contained several types of photoreceptor, their target *Vsx*+ interneurons and projection neurons.

Our expression comparisons between chicken retinal cell types and the *E. berryi* optic lobe (Fig. 4a) and retinal cell types (Extended Data Fig. 6a) reveal associations between the cell types and both ciliary and rhabdomeric photoreceptors, supporting both models. While the majority of *E. berryi* optic lobe neurons are more similar to rhabdomeric photoreceptors, some dopaminergic populations (Dopaminergic_1,3) display some similarity to vertebrate ciliary photoreceptors and bipolar cells (Fig. 5a and Extended Data Fig. 6e). This observation suggests that the cell types of the cephalopod visual system, similarly to vertebrates, may have been assembled by building upon both ciliary and rhabdomeric photoreceptors lineages. Moreover, in line with the proposed conservation of a conserved *Vsx*+ interneuron module (model 2), we also observed specific expression of orthologues of general vertebrate bipolar cell markers (*Vsx2*, *Otx2* and *Lhx3*) in the squid optic lobe clusters Glutamatergic_1 and Cholinergic_1 (Fig. 5b). Notably, cone bipolar off-centre cell type marker orthologues were enriched in Cholinergic_1 of *E. berryi*. These cholinergic cells are associated with *Aces-2*+ and *Nkx21*+ whose expression was localized to the medulla (Fig. 3g) and may

correspond to the 'bipolar cells of the medulla' described by J. Z. Young. We note that this prediction is based on gene expression similarity, and we do not imply that vertebrate and cephalopod bipolar cell types have a similar role in the visual signal processing circuitry.

### Contribution of gene evolutionary history to convergence

Distinct cell types of *E. berryi* are distinguished by paralogous genes: photoreceptors with distinct r-opsin paralogues and optic lobe dopaminergic cells with different *Grik2* receptors. We evaluated the role of gene duplication in the emergence of cell type diversity by pinpointing novel and duplicated genes among cell type markers (Wilcoxon rank sum test; Supplementary Table 7). Most retina and optic lobe neuronal cell types markers belong to pan-bilaterian gene families (Fig. 5c,d and Extended Data Fig. 9a,b) whereas non-neuronal cell types such as blood, white body, glial and progenitor populations have markers of more recent phylogenetic origin, with the exception of retinal photoreceptors that possess cephalopod-specific markers (Fisher enrichment test; Extended Data Fig. 9a).

While many gene families have pre-bilaterian origins, most cell-type-specific genes (2,342/3,759; 62.3%) have undergone duplication events at more recent nodes (cephalopod, spiralians and molluscs). We found that many dopaminergic and cholinergic neuronal cell types of the optic lobe are significantly enriched in cephalopods paralogues (Fig. 5h and Supplementary Table 10) contributing to distinct cell types (Extended Data Fig. 9c). Cholinergic_1 features cephalopod-duplicated potassium channels (*Kcna-1* and *Kcnn-1*), acetylcholine receptors (*Ach1-3* and *Acha4-3*) or ionotropic glutamate receptor (*Grik2-2* and *Grik2-4*) (Supplementary Tables 10). In the retina, we observe fewer expressed lineage-specific gene duplications, except in lentigenic cells expressing duplicated σ-crystallins. These findings emphasize the role of paralogues neo- and/or subfunctionalization of distinct phylogenetic origins in establishing the diversity of cephalopod neural cell types.

Previously reported gene duplications appear to play a lesser role than anticipated in the establishment of cephalopod cell type diversity[20]. C2H2 TFs (737 copies in *E. berryi*) are found in non-neuronal (endothelial and immune) and neuronal cells, but are not expressed in a cell-type-specific manner in hatchlings and only prominently expressed in optic lobe differentiating neurons (Neuro1; Extended Data Fig. 8b,e). Protocadherins (Pcdhs, 320 copies in *E. berryi*) could be subdivided into ubiquitously expressed and specifically expressed genes in the adult (expressed in roughly half the cell types of the optic lobe) with a different subset of Pcdhs expressed in differentiating or glial cells in hatchlings and perioesophageal brain (Extended Data Fig. 8a–d). Conversely, cholinergic neurotransmission plays an essential role in the cephalopod optic lobes as indicated by the expression of cholinergic receptor subunits in multiple cell types. A specific 'code' of metabotropic receptors and ionotropic receptor subunits designate different cells that receive cholinergic signals (Fig. 5i). Often, subunits originated through spiralian and bilaterian gene duplications are expressed in the same cell clusters, suggesting they could take part in hetero-pentamers. The cholinergic and glutamatergic receptor codes (Figs. 4f,g and 5i) are more prominent in adults than in hatchlings, consistent with an overall maturation of the optic lobe.

### Discussion

In contrast to vertebrates, the squid retina is remarkably simple, with only photoreceptors and support cells but no interneurons. Even so, we found unexpected complexity with two previously uncharacterized photoreceptor subtypes distinguished by two rhabdomeric opsins. While potentially a peculiarity of *E. berryi*, these cell types are not necessarily related to colour vision and could participate in polarized light detection[77–80]. Like ciliary photoreceptors of vertebrates, rhabdomeric photoreceptors of *E. berryi* are glutamatergic, suggesting this could be an ancestral condition in bilaterians.

The striking complexity of cell types in the optic lobe (26 cell clusters) raises the question of how this structure evolved from the last common bilaterian ancestor who probably had a rather small number of cells dedicated to visual signal processing[11]. Our comparative analyses support one evolutionary scenario inspired by analyses of the vertebrate retina[8,65] and the *Drosophila* optic lobe[66,81]: the ancestor of bilaterians possessed a mixture of rhabdomeric and ciliary photoreceptors, and two ancestral neuronal cell types corresponding to (1) a first-order interneuron, giving rise to vertebrate bipolar cells, transmedullary fly neurons and bobtail squid dopaminergic 1–3 neurons, and to (2) a second-order interneurons, giving rise to vertebrate ganglion, amacrine and horizontal cells, fly lobula complex neurons and bobtail squid cholinergic 1 and glutamatergic 1 neurons (and possibly others)[11].

We identified cell types corresponding to hypothesized centrifugal cells described by Cajal[4] and Young[6], which make retrograde projections from optic lobes to the retina: two dopaminergic cell populations in the medulla expressing *Pax6* and FMRF-amide transcripts. The FMRF-amide neuropeptide, but not its transcript, can be detected in the retina (Fig. 3c), suggesting that FMRF may be the retrograde signal from optic lobe to retina, consistent with recent findings in cuttlefish[54]. Retinal photoreceptors also express Dopr1 dopamine receptors, suggesting another potential retrograde signalling pathway, consistent with dopamine-induced photoreceptor pigment migration[82]. The function of these retrograde signals in cephalopods is unknown but may be analogous to retrograde signals from horizontal cells to photoreceptors that mediate lateral inhibition in vertebrates[55]. Alternatively, retrograde signals may modulate photoreceptor sensitivity[83].

Optic lobe neurons express a cell-type-specific code of glutamatergic and cholinergic receptors derived from cephalopod-specific duplications (Figs. 4f and 5i). Optic lobe neurons are predominantly dopaminergic unlike vertebrate or *Drosophila* brains. It is curious that the glutamate–dopamine co-neurotransmission, previously thought to be limited to mammalian ventral striatum, zebrafish olfactory lobes and *Drosophila* mushroom bodies[53], appears wide-spread in dopaminergic neurons of cephalopods. The cephalopod nervous system grows considerably over the life of the animal (unlike analogous structures in flies and mammals), and this maturation is accompanied by structural changes and the usage of different neurotransmitters in mature adult and hatchling cell type complements (Figs. 3a and 4a and Extended Data Fig. 6b–e). These distinct ongoing patterns of differentiation in the cephalopod nervous system are reminiscent of the critical period found in mammals.

Comparison among recently published single-cell datasets in other cephalopods[84–86] indicated that the diversification of dopaminergic neuronal cell types is a global signature of cephalopod optic lobes, as well as the molecular conservation of the FMRF+ centrifugal and putative cephalopod bipolar cells as well as glial markers (*Eaa1-2* and *Glna-2*; Extended Data Fig. 7 and Supplementary Notes 2 and 3). Independent diversification of dopaminergic cell types in decapods and octopods could be interpreted as adaptations to their unique habitats and evolutionary histories.

While the same gene families are expressed in bobtail squid, vertebrate and fly visual systems (Fig. 5c,d), many of these genes underwent duplication events in the lineage leading to the bobtail squid and prominently in the cephalopod lineage (Fig. 5e–h). Paralogues thus probably played a key role in the elaboration of novel cell types as recently described in euteleosts[87]. Extensive chromosomal rearrangements (Fig. 1e) have been linked with the acquisition of novel gene regulatory mechanisms at the base of vertebrates[88] and recently in the cephalopod lineage[89], which could have permitted the acquisition of cell-type-specific expression for gene duplicates. The exact role of such mechanisms could be addressed by profiling the visual system cells of a spiralian that has retained the ancestral chromosomal architecture, such as the sea scallop, to understand how these spiralian duplicates acquired their expression domains in cephalopods (Fig. 1e–g).

## Methods

### Ethics statement

This study was carried out in accordance with procedures authorized by Guidelines for Proper Conduct of Animal Experiments by the Science Council of Japan[90]. Despite the absence of legislation pertaining specifically to cephalopods in Japan, we aspired to abide by the highest standards in the field. All conducted experiments were therefore also in line with EU Directive 2010/63/EU and with the guidelines and the principles detailed in refs. 91–94. All experiments were approved by the Okinawa Institute of Science and Technology Graduate University Animal Care and Use Committee (approval ID: 2018-204). No transgenic animals were used in this study.

### Animal care

Adult *E. berryi* were collected from the coast of Mie prefecture in Japan and transported to Okinawa where they were acclimated to temperature (20 °C) and pH (8.3) of a closed aquarium system in filtered natural seawater obtained from the shores of Okinawa, Japan (OIST Seragaki Marine Science Station). Animals were maintained essentially as described previously[17] until they were euthanized for experiments. Animals were exposed to a static 12 h:12 h light:dark cycle. Tanks that housed the animals contained an enriched environment including natural substrate (autoclaved sand or crushed coral), parts of clay pots and natural rocks as dens. Mature animals were fed daily with opossum shrimp or mysids, whereas *Neomysis japonica* proved a suitable prey for hatchlings. Fresh glass shrimp, *Palaemonetes* spp. and frozen shrimp that were purchased in local grocery stores were fed to late juveniles and adults. Tanks were cleaned daily to remove uneaten food and waste matter. Before experiments, animals were euthanized using 4% ethanol in sterile-filtered natural seawater. Animals were allowed to breed freely. Hatchlings were obtained either from eggs provided by females impregnated in the wild or by breeding wild animals in the laboratory.

### Genomic DNA extraction and library construction

Germline genomic DNA was obtained from the mature testes of a single mature *E. berryi* male individual by OIST Sequencing Section (SQC). After thorough grinding in liquid nitrogen, cell lysate was embedded in low-melting agarose plugs, subjected to proteinase K digestion, washed in 20 mM Tris, 50 mM EDTA, pH 8.0 and released using agarase. DNA molecular size was assessed using a FemtoPulse instrument (Agilent) and found to be made of fragments >50 kb suitable for continuous long-read sequencing. We used several sequencing technologies to assemble the genome. First, two libraries of paired-end 250-bp-long Illumina libraries (with insert sizes of 350 bp and 800 bp) were prepared and sequenced using HiSeq2500 in Rapid Run mode yielding 115 Gb and 120 Gb providing 19× and 21× coverage, respectively, at OIST SQC. Demultiplexing was carried out using bcl2fastq (v.2.19). Long-insert Pacific Bioscience continuous long-read sequencing libraries were prepared using v7 chemistry, and 16 single molecule sequencing in real time (SMRT) cells were sequenced on a Sequel instrument at the Vincent J. Coates Genomics Sequencing Laboratory at UC Berkeley. Pacbio sequencing yielded 8.8 million reads representing 160.7 Gb of data with a median read length of 14,000 and read N50 of 32 kb. Finally, we generated 10× linked reads using the Chromium system (10x Genomics) and sequenced them on two lanes of Hiseq4000 in 2x150bp mode[95]. Demultiplexing was achieved using Supernova (v.2.1.1). Chromatin contact information was obtained from an optic lobe sampled from an adult individual crosslinked in 1% paraformaldehyde (PFA). The HiC library was constructed and the genome was scaffolded using HiRiSE software (Dovetail Genomics)[96] and was sequenced on a HiSeq4000 and NovaSeq6000 SP. Raw sequencing data were submitted to the National Center for Biotechnology Information under accession number PRJEB52690.

### Genome size estimation

The size of the genome was estimated using a *k*-mer spectrum approach as described in ref. 97 from 17-mer to 31-mer distribution calculated by Jellyfish (v.2.2.7)[98]. By performing a sum of unique *k*-mers weighted by their multiplicity, we originally found an effective haploid genome size of 5.6 Gb.

### Genome assembly and validation

We used wtdbg2 (v.2.5) with default parameters and genome size parameter '-g 4.6 G' to assemble the Pacific Bioscience reads[99]. To generate consensus sequence contigs, Pacific Bioscience reads were mapped with minimap2 (v.2.16)[100] and to the obtained contig assembly in two rounds using the wtdbg2 polishing module. We also utilized the Illumina paired-end date in an additional polishing step using Racon (v.1.3.2)[101], resulting in a haploid reference of 5.9 Gb composed of 51,130 contigs and an N50 of 831,242 bp. We further scaffolded contigs with scaff10x (https://github.com/wtsi-hpag/Scaff10X) with parameters '-longread 1 -gap 100 -matrix 2000 -read-s1 10 -read-s2 10 -score 20 -edge 50000 -link-s1 10 -link-s2 10 -block 50000'. We assessed the completeness and haplotype collapse of the assembled genome using BUSCO (v3.1.0)[102], yielding C:84.9% [S:82.7%,D:2.2%], F:5.8%,M:9.3% n:978 (Supplementary Fig. 1). We also verified the homogeneity of base coverage in final contigs by remapping both the Pacbio using minimap2 (ref. 100) and Illumina datasets using Burrows–Wheeler Aligner (BWA)[103] and analysing the binary alignment/map (BAM) file with the alfred tool[104] (https://github.com/tobiasrausch/alfred). BAM files were sorted using Samtools (v.1.9)[105].

To further extend the contiguity of our haploid reference, we used long-range contact information from HiC. The assembly was scaffolded using the HiRise pipeline by Dovetail Genomics. To validate the output assembly, HiC data were processed using Juicer (v.e0d1bb7) and contact density was examined and inspected using Juicebox software[106]. Our final assembly shows 48 main chromosomes (length >1 Mb; Supplementary Fig. 1a,b). The number of chromosomes is consistent with the reported diploid number of chromosomes 96 in decapods[107,108]. Additional quality control consisted of remapping Illumina RNA-seq data and Pacific Biosciences Iso-seq reads (see below). Bulk RNA-seq Illumina reads contained on average 86.58% of reads mapping uniquely to the genome. Mapping Iso-seq reads yielded 50,400 aligned (99.51%), of which 45,243 on 100% of their length, 47,379 at 95% and 48,829 at 75% (that is, 97% of transcripts).

### Bulk RNA extraction and sequencing

We obtained total RNA from embryos over embryonic developmental (stages 0–28) and from 26 samples of organs of a >100-day-old adult male. Total RNA extraction was carried out using TRIzol reagent (Invitrogen) for lysis followed by on-column purification using PureLink RNA Mini Kit (Invitrogen) as described in the TRIzol Plus RNA Purification instructions from manufacturer. In brief, tissues were dissected and directly homogenized in 1 ml of TRIzol reagent on ice. After 5 min incubation at room temperature, 0.2 ml of chloroform was added, vortexed and incubated for an additional 3 min at room temperature. Phase separation was achieved by centrifugation at 12,000*g* at 4 °C. The upper colourless phase was mixed with an equivalent volume of 70% ethanol in DEPC-treated water, mixed, vortexed and applied directly to the PureLink RNA Mini Kit columns. Wash steps were followed as per the manufacturer's protocol. RNA extraction was followed by Turbo DNase treatment (Thermo Fisher Scientific) to remove genomic DNA according to the manufacturer's instructions. RNA quality was evaluated using TapeStation 4200 BR RNA Screentape and reagents (Agilent Technologies). For organ-specific transcriptomes, library preparation was achieved by using TruSeq Stranded mRNA Library Prep for NeoPrepTM (Illumina). For stage-specific transcriptomes, we used NEBNExt Ultra II Directional RNA Library Prep Kit for Illumina (NEB). Organ-specific RNA-seq was carried out on a HiSeq4000 and embryonic-stage-specific RNA-seq was done on a Novaseq6000 SP, yielding a total of

1,966,727,071 reads. Demultiplexing was carried out using bcl2fastq (v2.19). Reads were mapped to the *E. berryi* genome using STAR v2.5.2 in several rounds[109]. In a first round, the following parameters were used: '--outSAMmapqUnique 255 --outFilterMultimapNmax 10 --outReadsUnmapped Fastx --outFilterMismatchNmax 999 --winBinNbits 10 --chimOutType SeparateSAMold --chimSegmentMin 20 --chimJunctionOverhangMin 20 --outFilterMismatchNoverLmax 0.5 --outFilterMismatchNoverReadLmax 10 --outSAMstrandField intronMotif --alignSoftClipAtReferenceEnds No --alignMatesGapMax 5000 --outFilterScoreMin 100'. In a second round, we utilized the junctions observed in the first round, using the following parameters: --outSAMstrandField intronMotif --outSAMtype BAM SortedByCoordinate --outSAMmapqUnique 255 --outFilterMultimapNmax 1 --outReadsUnmapped Fastx --chimJunctionOverhangMin 20 --alignSoftClipAtReferenceEnds No --sjdbFileChrStartEnd *SJ.out.tab --limitSjdbInsertNsj=5000000 --sjdbInsertSave All'. Count tables were obtained using subreads FeatureCounts[110] using '-p -B -t exon -g gene_id' parameters. We also sequenced full-length cDNA from retinas and optic lobes on a Sequel instrument. Following the Iso-seq protocol, circular consensus (CCS) of subreads was calculated, and subsequently classified as full-length or non-full-length based on the presence of SMART adaptors at both extremities. Full-length transcripts were clustered and polished using all CCS reads with quiver (https://github.com/ben-lerch/IsoSeq-3.0).

### Genome annotation

We generated RNA-seq data for adult organs and developmental time series with 1,966,727,071 reads (Supplementary Tables 2 and 3). We aligned the reads to the genome using STAR (v2.5.2b) and reached >88% uniquely mapping reads[109]. These alignments were subsequently used to assemble transcriptomes for each organ using Stringtie (v1.3.3b)[111–113] and then merged together using Taco[114]. In parallel, a de novo assembly of all RNA-seq was performed using Trinity[115]. Assembled transcripts from de novo and genome-guided Trinity, as well as the high-quality Iso-seq transcripts, were aligned to the genome using GMAP (version of 2019-02-26)[116] using parameters '-f 3 -n 0 -x 50 -t 16 -B 5 --gff3-add-separators=0 --intronlength=500000'. Mikado (v1.2.1)[117] was used to generate a high-quality reference transcriptome leveraging (1) the trinity transcriptomes, (2) the Iso-seq transcripts and (3) the Stringtie transcriptomes merged with Taco and a set of curated splice junctions generated from RNA-seq alignments using Portcullis (v1.0.2) using parameters 'full --strandedness firststrand --bam_filter'[118]. Putative fusion transcripts were detected by Blast comparison against Swiss-Prot, and open reading frames were annotated using Trans-decoder (https://github.com/TransDecoder/TransDecoder). Transcripts derived from the reference transcriptome were selected to train Augustus (v.3.3.3) de novo gene prediction tool[119]. Exon positions in the Mikado transcriptome assembly were converted into hints for Augustus gene prediction. We aligned the proteomes of cephalopod species bobtail *Euprymna scolopes*[119], octopod *Octopus bimaculoides* and squid *Doryteuthis pealleii*[19] and provided them as CoDing Sequence (CDS) hints for Augustus gene prediction.

Finally, a repeat library was constructed using RepeatModeler (v.1.0.11)[120] and used for masking with RepeatMasker (v.4.0.7)[121]. Gene models with more than half or more of their exons overlapping with more than half of those of repeats were discarded, yielding 56,767 filtered gene models. Alternative transcripts and untranslated regions (UTRs) were subsequently incorporated using the PASA (v.2.41) pipeline[122]. PFAM domains were identified using PfamScan (v.1.6) The obtained gene models contain a total number of 5,277 distinct PFAM domains (SuppData-PFAM domains).

### Gene family reconstruction and phylogenetic analyses

We reconstructed gene families using Orthofinder (v2.33) using the non-redundant proteomes of selected species ranging across the tree of bilaterian and including some key model species for neurobiology work such as fruit fly, mouse and zebrafish[123]. We recovered 21,332 gene families, of which 12,046 contained at least one *E. berryi* gene family member.

### Dissociation of tissues into single cells

Tissues were dissociated using a 1% papain solution in filtered natural seawater containing 0.1% sodium thioglycolate for retinas, and a 1% pronase solution for non-retinal tissues on a nutator with gentle trituration. Cell clumps were filtered out using either 30-µm MACS SmartStrainers (Miltenyi Biotec) for v2 sample or using 40-µm and 70-µm cell strainers (flowmi). Cells were resuspended in a 50% L15 medium in sterile-filtered natural seawater. The single-cell suspension quality and cell concentration were evaluated using a C-Chip Disposable haemocytometer (NanoEntek) on a Zeiss PrimoVert Monitor Inverted microscope. The number of viable cells was evaluated using trypan blue (Thermo Fisher).

### Single-cell library preparation and sequencing

scRNA-seq library preparation was performed immediately after dissociation with the 10x Chromium v2 or v3.1 kits (10x Genomics) following the manufacturer's protocol. We produced five replicates of optic lobe scRNA-seq libraries from *E. berryi* that had hatched the same day (1 dph) and four replicates of optic lobe scRNA-seq libraries from mature adult *E. berryi* (1× >80 dph, 1× 60 dph and 2× 80 dph). We were unable to produce libraries derived from retinas of 1-dph hatchlings and used six replicates of adult individuals (2× 60 dph, 4× 80 dph and 1× >80 dph). We dissected the perioesophageal brain without the optic lobes to use as a control from mature adult *E. berryi* and generated five independent libraries (2× 60 dph, 2× 80 dph and 1× >80 dph) (Supplementary Table 4 and Supplementary Fig. 1). scRNA-seq was carried out on an Illumina HiSeq2000, HiSeq4000 or NovaSeq6000 instrument at OIST SQC. The samples obtained from >80-dph-old individual produced using the v2 kit were originally sequenced on a HiSeq4000 paired-end 150-bp run with scattered PhiX spikes ranging from 0% to 40% on each lane to reduce base bias. Raw sequencing data were processed with the CellRanger version 2.10 and subsequently v.3.1.0 mkfastq script (10x Genomics). Resequencing of the same libraries was achieved using an Illumina Novaseq6000 instrument using paired-end 150 bp with 10% PhiX spike and was similarly demultiplexed using CellRanger mkfastq v.3.1.0. The paired-end 150-bp-long fastq files were trimmed by quality and for size using sickle (https://github.com/najoshi/sickle). Read quality was examined using fastqc (v.0.11.5). All the other single-cell libraries were sequenced using Read1:26 bp, Index1:8 bp and Read2:98 bp, and demultiplexing was achieved using bcl2fastq (v.2.19). Hatchling (1 dph) optic lobe samples were sequenced on a Novaseq6000 instrument (Illumina) using a 10% PhiX spike.

### scRNA-seq read alignment and analysis

For scRNA-seq mapping only, all gene models were extended by 3 kb in the 3' direction using the script https://github.com/JulienPPichon/UTR_extension_GTF. Reads were mapped to the genome with STAR Solo[109,124] (Extended Data Fig. 2). A total of 6,709,213,691 reads were sequenced, of which 5,510,308,930 mapped uniquely (corresponding to 82% of all sequenced reads). STAR-Solo-based gene-barcode counts matrices were analysed with the Seurat (v.4.1.0) R package[125]. Numbers of features, unique molecular identifiers and percentage of reads mapping to the mitochondrial genome per cell were examined for each dataset (Extended Data Fig. 2a–d). Features found in fewer than three cells were removed from subsequent analysis. Cells in which more than 30% of reads corresponded to genes encoded by the mitochondrial genome were removed from subsequent analysis. A minimal number of cells expressing each feature was set for each dataset (Supplementary Table 4) ranging from 100 to 600 features per cell. Replicates were normalized individually using the 'SCTransform()' function before integration, using 'PrepSCTIntegration()' and

'FindIntegrationAnchors()' with 'normalization.method' set to 'SCT'. Data integration was achieved using the 'IntegrateData' command. Principal component analysis was performed using the 'RunPCA()' command using 1:200 principal components at first. The 'FindNeighbors()' and 'FindClusters()' commands were used to identify putative clusters at resolutions of 0.1, 0.2, 0.5, 0.75, 1, 2, 5 and 7. The number of relevant principal components was determined using the 'ElbowPlot' function of Seurat and by examining gene expression in UMAPs created using principal components from 1:10 to 1:200. We used the Seurat's 'FindAllMarkers' command and R package presto's 'wilcoxauc' command to identify differentially expressed genes in each cluster. First, using only the >80-dph and 60-dph samples of the retinal datasets, using 200 principal components with a resolution of 0.1, a group of cells that clustered with all other clusters across resolutions was observed ('cluster6' at a resolution of 0.1, corresponding to 74 cells) and was removed as this behaviour was reminiscent of cell doublets. Second, after the other retinal datasets were generated and reclustered using 1:35 principal components and an additional group of cells that expressed markers of both neurons and haemocytes (cluster 2 at a resolution of 0.2, corresponding to 2,488 cells) was deemed to correspond to cell doublets and removed from subsequent analysis. For the retina dataset, we utilized higher-resolution cell clusters for neuronal and retinal clusters using resolution 0.5 for the photoreceptors, pigmented-tyro⁺, pigmented2, white body and lentigenic-associated cell populations, and resolution 0.2 for lentigenic cells, but a lower resolution of 0.1 for clusters representing cell types that are not specific to the retina (haemocytes, endothelium and fibroblasts), as our interest does not lie in these cell types. For the adult optic lobe dataset, we used 200 principal components and a resolution of 0.75. For the hatchling optic lobe dataset, we used 200 principal components and a resolution of 1. For non-optic lobe brain control datasets, we used a resolution of 3. Markers of each cluster for each dataset are available in Supplementary Tables 4–7.

## X-ray microtomography (micro-CT)

Euthanized animals were fixed in 4% PFA in seawater for at least overnight at 4 °C. Samples were dehydrated serially in 25%, 50%, 75% and 100% ethanol at 4 °C. Samples were kept in each concentration of ethanol overnight and then transferred to 1% iodine solution (1.9 g iodine per 50 ml of absolute ethanol) at room temperature for at least 24 h (or longer for larger samples). After staining, samples were washed in absolute ethanol three times.

X-ray microscope Xradia 510 Carl (Zeiss) was used to obtain micro-CT images of squid samples. Small hatchlings were mounted in heat-sealed pipette tips or 0.2-ml PCR tubes. Older samples were mounted in 1.5-ml Eppendorf tubes. In the containers used during image acquisition, samples were either submerged in ethanol or placed on Kim wipe or plastic wrap soaked in ethanol. Kim wipe and saran wrap were also used to prevent samples from moving within the container. We used standard objectives (0.39×, 4×) and different combinations of source–sample and sample–detector distances to fit and optimize the entire region of interest. Images were acquired with an exposure time of 1–2 s per projection and a total of 1,601 projections over 360°. No filter was used. The vertical stitching feature was applied to join multiple tomographies.

The projection data from each scan were reconstructed using the integrated volume reconstruction software of the Xradia machine. The resulting reconstructions were exported in .txm or DICOM format and imported into Amira (version 6.5, Thermo Fisher Scientific) for manual segmentation of anatomical structures and to render isosurfaces. For central nervous system brain lobes, we segmented the inner, lobular neuropil without its confluent outer perikaryal layer. The neuropil layer in the centre of lobes can be seen as darker shades of grey in tomography, while the surrounding perikaryal layer is seen as lighter shades of grey.

The surface generation algorithm implemented in Amira (Generate Surface module) was applied for each segmented material using smoothing function values <2.5. The resulting three-dimensional surface meshes were exported in .ply format and imported into a ParaView package for data visualization[126].

## Transmission electron microscopy

Optic lobe and retinal samples dissected from freshly euthanized animals were fixed in 4% PFA in filtered seawater overnight at 4 °C. Samples were then stored in 2.5% glutaraldehyde in 1× phosphate-buffered saline (PBS) for up to 1 month at 4 °C. To enhance contrast of the samples for transmission electron microscopy observation, samples were stained with 2% osmium tetroxide in water for 1 h and then washed with deionized water five times for 3 min each. Samples were then blocked and stained in 4% uranyl acetate overnight at 4 °C and washed with deionized water four times for 3 min each. Samples were then serially dehydrated in 30% and 70% acetone diluted in water for 15 min each. Samples were then further dehydrated in 100% acetone for 15 min.

The 15-min dehydration step in fresh 100% acetone was repeated two additional times. For embedding samples, epoxy consisting of 20 ml Epon 812 Resin, 10 ml dodecenyl succinic anhydride (DDSA), 10 ml methyl-5-norbornene-2,3-dicarboxylic anhydrid (MNA) as hardener and 1 ml 2,4,6-tris(dimethylamino)phenol (DMP30) were mixed in a 50-ml conical tube and then rotated in a rotor for 1 h before use. Samples were first incubated in 33% and then 77% epoxy diluted in 100% acetone for 1 h each, and then transferred into 100% Epon for 1 h incubation. Samples were then transferred into fresh 100% Epon for overnight incubation. All steps that are part of the acetone dehydration and Epon incubation up to this stage were performed at room temperature.

To embed samples in epoxy, samples were placed in a silicon mould filled with epoxy resin Epon formula (Taab Company) and then incubated at 60 °C for 2 days in an oven. After polymerization of the epoxy, blocks with samples were sliced into 70-nm sections using a Leica EM UC7 Ultramicrotome and placed on grids (Nisshin). For sectioning, a diamond knife (Diatome) was used. Before observation under a transmission microscope, grids with sections were restained with 4% uranyl acetate for 20 min and then washed with deionized water three times for 3 min each. Image acquisition was performed with a transmission electron microscope (JEOL JEM-1230R, 100 KV TEM).

## Hybridization chain reaction

*E. berryi* were washed several times in natural seawater, euthanized in 4% ethanol in sterile-filtered natural seawater and fixed in 4% formaldehyde in sterile-filtered natural seawater overnight at 4 °C. After three washes in 1× PBS, the squid were dehydrated in increasing concentrations (25%:75%, 50%:50%, 75%:25% and 100%:0%) of methanol:PBS for 15 min each and stored in 100% methanol at −30 °C until necessary. For staining, individuals were rehydrated in a reversed-order methanol-to-PBS series (25%:75%, 50%:50% and 75%:25%). For whole-mount stainings, rehydrated individuals were utilized directly for HCR. When cryosections were required, fixed samples were first incubated in 25% sucrose in PBS solution overnight and subsequently in a 35% sucrose in PBS solution overnight or until the sample sank to the bottom of the Falcon tube. After sucrose incubation, samples were embedded in optimal cutting temperature compound (OCT) cryosection medium and cryosectioning was performed using either a CM3050S or CM1950 cryostat (Leica). HCR probes (version 3 chemistry) were designed using the Molecular Instruments website (www.molecularinstruments.com). RNA sequences utilized for probe design are presented in Supplementary Table 3. In brief, section HCR staining was performed as follows: slides were rinsed in chilled 1× PBS for 1 h at 4 °C to remove OCT, then the samples were enclosed with iSpacer (0.5 mm, SUNJin Lab) and surrounded inside the iSpacer square with a pap pen. Slides were transferred to a humidified chamber at 37 °C and treated with 10 μg ml⁻¹ Proteinase K in PBST for 10 min to permeabilize the cell membrane, followed by

two rinses with 2 mg ml⁻¹ glycine in PBST at room temperature and post-fixation with 4% PFA for 30 min. Sections were washed with PBST for 5 min three times before prehybridization. Each sample was prehybridized by adding 200 µl of hybridization buffer and incubated at 37 °C for 10 min inside the humidified chamber. Hybridization was carried out by exchanging the hybridization buffer with 0.4 pmol of the probe set, followed by overnight incubation at 37 °C. After overnight incubation, excess probes were sequentially rinsed in probe wash buffer to which 5× saline sodium citrate buffer with Tween (SSCT) had been added to final concentrations (vol/vol) of 0%, 25%, 50%, 75% and then 100% for 15 min each at 37 °C. Slides were rinsed before preamplification with 5× SSCT for 5 min and incubated in amplification buffer for 30 min at room temperature. After removing the preamplification buffer, 100 µl of hairpin solution, prepared according to the manufacturer's instructions and snap-cooled, was added to each sample and incubated in a dark humidified chamber at room temperature overnight. Excess hairpin solution was removed through two rinses with 5× SSCT for 30 min, followed by a final rinse for 5 min. Finally, ProLong Gold Antifade Mountant (ThermoFisher) was added to the samples, and sections were sealed with coverslip for imaging. Whole-mount HCR was carried out by following the manufacturer's protocol (whole-mount sea urchin embryos' protocol), with an additional clearing step using the RapiClear solution (SUNJin Lab). Antibodies were diluted at 1:100 (Ac-Tubilin Ab24610) or 1:1,000 (anti-FMRF Ab15348), and 50 µl was added to samples. Samples were mounted in an iSpacer (1 mm, SUNJin Lab) filled with RapiClear solution. Staining results were visualized using LSM780/LSM880 at OIST Imaging facility or LSM980 at UCL Imaging Facility. Images were analysed using ZEN software and Fiji.

### Cross-species comparisons of single-cell data

Dataset count tables and embeddings were downloaded from https://singlecell.broadinstitute.org/single_cell for datasets listed in Supplementary Table 12. Seurat objects were generated according to the embedded cell annotation. Using our OrthoFinder gene families (see 'Gene family reconstruction and phylogenetic analyses' section), we reduced datasets to the genes belonging to shared gene families for both species compared as follows: (1) we filtered out genes without expression in scRNA-seq datasets, (2) we retained all genes with 1-to-1 homology after step 1 filtering and (3) we arbitrarily selected a single gene for each species for remaining many-to-many homologues. This procedure allows the use of MetaNeighbor for cross-species comparisons[69,70] and limits the impact of distant out-paralogues on computed similarity scores. After identifying variable genes, MetaNeighbor area under the receiver operating characteristic curve (AUROC) scores were calculated using the Unsupervised metaneighbor function in fast mode (default parameters). To assess consistency of the MetaNeighbor results, we also used SAMap[71] with default parameters.

### Reporting summary

Further information on research design is available in the Nature Portfolio Reporting Summary linked to this article.

## Data availability

Raw genomic DNA sequencing data and genome assembly generated during this study are available through European Nucleotide Archive (ENA) accession number PRJEB52690. Raw and processed bulk-RNA seq and scRNA-seq data are available through the National Center for Biotechnology Information (NCBI) Gene Expression Omnibus (GEO) accession number GSE203527. We provide an interactive web-based Shiny app (http://141.164.60.190:3838/Eberryi_visual/) to explore the single-cell datasets.

## Code availability

Code used for analysis is available via GitHub at https://github.com/dariagavr/EberryiVisual (ref. 127).

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

## Acknowledgements

We acknowledge the OIST sequencing facility 'SQC' and its members for their support: N. Arakaki, S. Yamasaki, M. Kawamitsu and T. B. H. Soliman as well as Vincent J. Coates Genomics Sequencing Laboratory at UC Berkeley and particularly S. McDevitt for Pacific Bioscience sequencing. The OIST HPC facility was used for computational analyses. We owe special thanks to T. Kakeya from Scrum Inc. for teaching us to perform scRNA-seq libraries using 10x Genomics reagents. We thank I. Masai for access to cryostat and Y. Nishiwaki for assistance and training to utilize the cryostat, and M. Hall and A. Takahashi from the OIST Imaging section for assistance with cryosectioning. We thank A. Greig from the UCL Imaging facility for expert advice on imaging and training and A. Sadier for advice on HCR stainings, and are grateful to T. Sasaki and M. Hall from the OIST Imaging Section for training and assistance with transmission electron microscopy. We thank the Evolutionary Neurobiology (Watanabe) Unit at OIST for the anti-Fmrf-amide antibody. We thank R. Nakajima for professional photography of *Euprymna berryi*, and J. Jolly and R. Kawaura for early contributions to bobtail squid aquaculture. We thank J. Pichon for optimizing the UTR extension Python script. We are grateful to C. Plessy, C. Martín-Durán, A. de Mendoza, M. Feller, K. Shekhar and M. Kuba for critical reading of the manuscript and helpful comments and suggestions. We are grateful to Z. Ziadi for insightful calculations based on micro-CT and optic lobe growth modelling. We are grateful to J. M. Morales for valuable comments. This work was supported by internal funds of the OIST Molecular Genetics Unit, the Chan Zuckerberg Biohub to D.S.R. D.G. was supported by a JSPS fellowship for prospective researchers (Fellow ID no. PE17011). D.S.R. is grateful for the support of the Marthella Foskett Brown Chair in Biology at UC Berkeley. E.P. was supported by a Newton International Fellowship from the Royal Society (NIF\R1\222125). F.M. was supported by a JSPS Kakenhi grant (grant no. 19K06620), BBSRC (grant no. BB/V01109X/1), Leverhulme Trust (grant no. RPG-2021-436) and Royal Society (grant no. URF\R1\191161) funding.

## Author contributions

D.G., F.M. and D.S.R. conceived the project. D.G. and F.M. assembled the genome, carried out bulk RNA-seq extractions, performed annotation, performed scRNA-seq and analysed data. D.G., Y.T. and Y.H. carried out histology and HCR and antibody stainings. F.Z.-K. Y.H. carried out micro-CT sample preparation. C.S., Y.H. and F.Z.-K. analysed micro-CT. L.Z. extracted haemocytes, prepared samples for histology and prepared cryosections. C.S. performed central nervous system dissections and interpreted HCR images. C.S. and Y.T. carried out photography. L.P. carried out imaging of HCR stainings and single-cell comparisons. E.P. performed single-cell analyses and contributed to revisions. N.L. contributed to staining, imaging efforts and supervision. D.G., F.M. and D.S.R. wrote the manuscript with input from all authors.

## Competing interests

D.S.R. is a shareholder and advisor to Cantata Bio. The other authors declare no competing interests.

## Additional information

**Extended data** is available for this paper at https://doi.org/10.1038/s41559-025-02720-9.

**Correspondence and requests for materials** should be addressed to Daria Gavriouchkina, Ferdinand Marlétaz or Daniel S. Rokhsar.

[1]UK Dementia Research Institute, University College London, London, UK. [2]Molecular Genetics Unit, Okinawa Institute of Science and Technology Graduate University, Onna, Japan. [3]Genomics and Regulatory Systems Unit, Okinawa Institute of Science and Technology Graduate University, Onna, Japan. [4]Centre for Life's Origins and Evolution, Department of Genetics, Evolution and Environment, University College London, London, UK. [5]Nonlinear and Non-equilibrium Physics Unit, Okinawa Institute of Science and Technology Graduate University, Onna, Japan. [6]Biomedical Neural Dynamics Collaboration Laboratory, Riken Center for Brain Science, Wako, Japan. [7]Department of Biology, Keio University, Yokohama, Japan. [8]DOE Joint Genome Institute, Lawrence Berkeley National Laboratory, Berkeley, CA, USA. [9]Department of Molecular and Cell Biology, University of California, Berkeley, CA, USA. [10]Chan-Zuckerberg BioHub, San Francisco, CA, USA. ✉e-mail: daria.gavriouchkina@gmail.com; f.marletaz@ucl.ac.uk; dsrokhsar@gmail.com

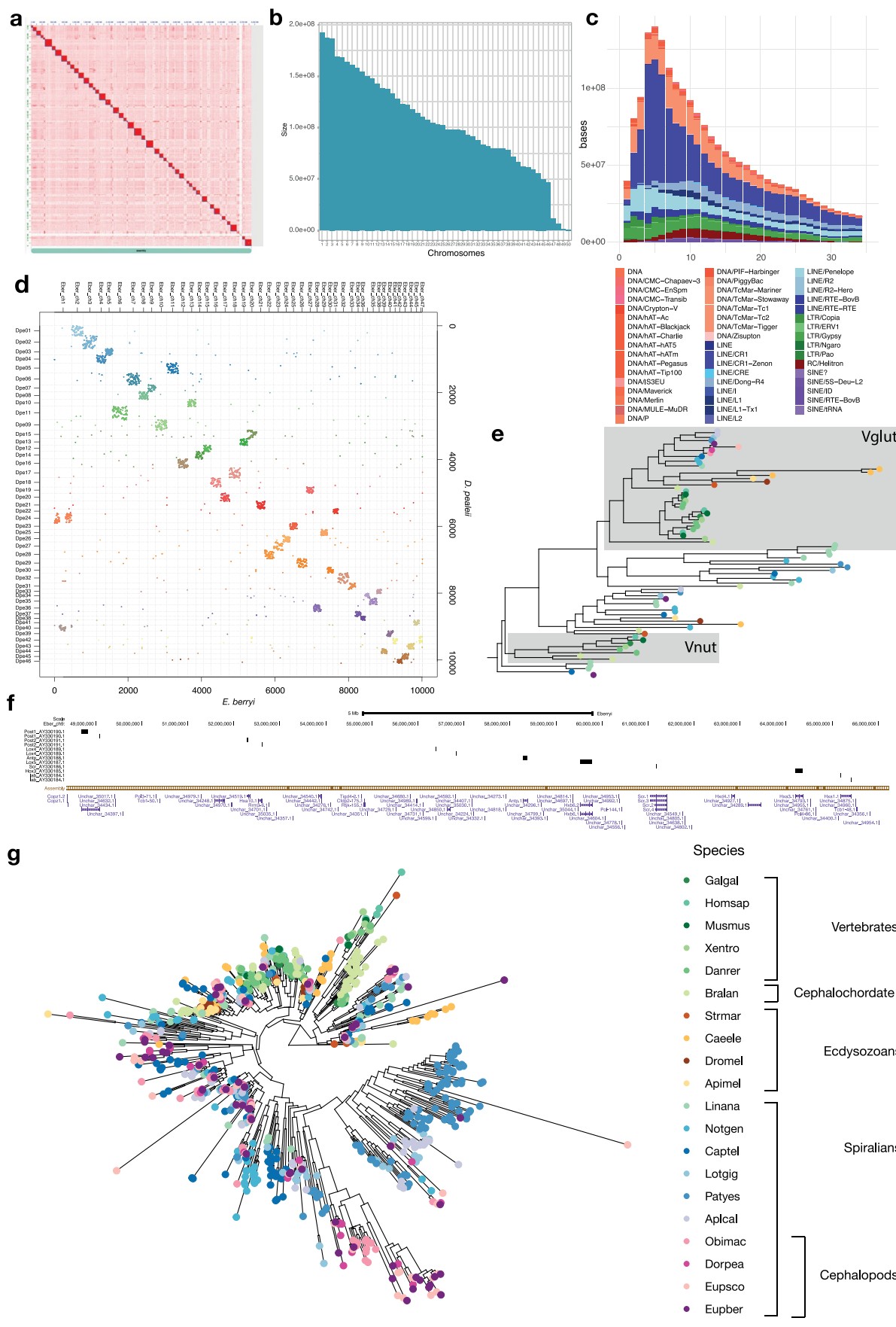

**Extended Data Fig. 1 | See next page for caption.**

**Extended Data Fig. 1 | *Euprymna berryi* Genome Assembly QC and Characterisation. a**, Image of HiC contact map. **b**, Barplot indicating number of chromosomes > 1 Mb sorted by descending length, that were deemed to correspond to a chromosome. **c**, Repeat landscape (Repeat modeler). **d**, Oxford plot comparing chromosome complement of *E. berryi* to that of *D.pealei*. **e**, Phylogenetic tree of Vglut and Vnut transporters. **f**, UCSC Genome Browser screenshot demonstrating full 17 Mb Hox cluster in *E. berryi*. **g**, Phylogenetic tree of ionotropic acetylcholine subunits across species. Species abbreviations used: Galgal - *Gallus gallus* (chicken), Homsap – *Homo sapiens* (humans), Musmus –

*Mus musculus* (mouse), Xentro – *Xenopus tropicalis* (Frog), Danrer – *Danio rerio* (zebrafish), Bralan – *Branchiostoma lanceolatum* (lancelet), Strmar - *Strigamia maritima* (centipede), Caeele – *Caenorhabditis elegens* (nematode), Dromel – *Drosophila melanogaster* (fruit fly), Apimel – *Apis mellifera* (bee), Linana - *Lingula anatina* (brachiopod), Notgen – *Notospermus geniculatus* (nemertean), Captel – *Capitella teleta* (annelid), Lotgig – *Lottia gigantea* (limpet), Patyes – *Patinopecten yessoensis* (scallop), Aplcal – *Aplysia californica* (see hare), Obimac – *Octopus bimaculoides* (octopus), Dorpea -*Doryteuthis peallei* (longfin inshore squid), Eupsco – *Euprymna scolopes* (Hawaiian bobtail squid), Eupber – *Euprymna berryi*.

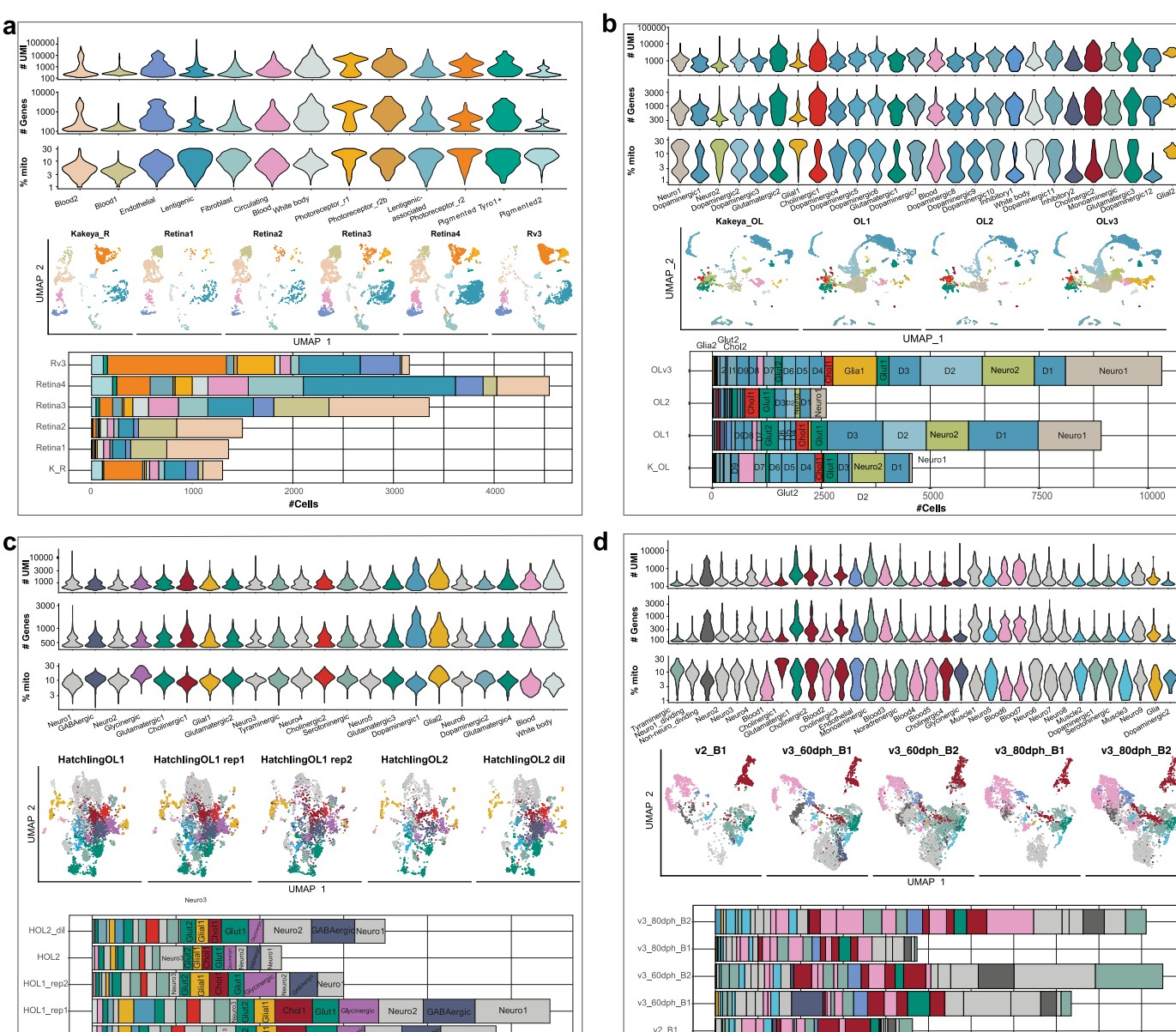

**Extended Data Fig. 2 | Quality control metrics of scRNA-seq. a-d,** QC Violin plots showing the number of UMIs (top row), genomic features (middle row) and percentage of genes encoded by the mitochondrial genome expressed in each cell cluster (bottom row), UMAP projections after integration and barplots indicating contribution in terms of numbers of cells from each replicate, for each dataset: **a**, retina **b**, adult optic lobe **c**, hatchling optic lobe **d**, non-optic lobe brain control.

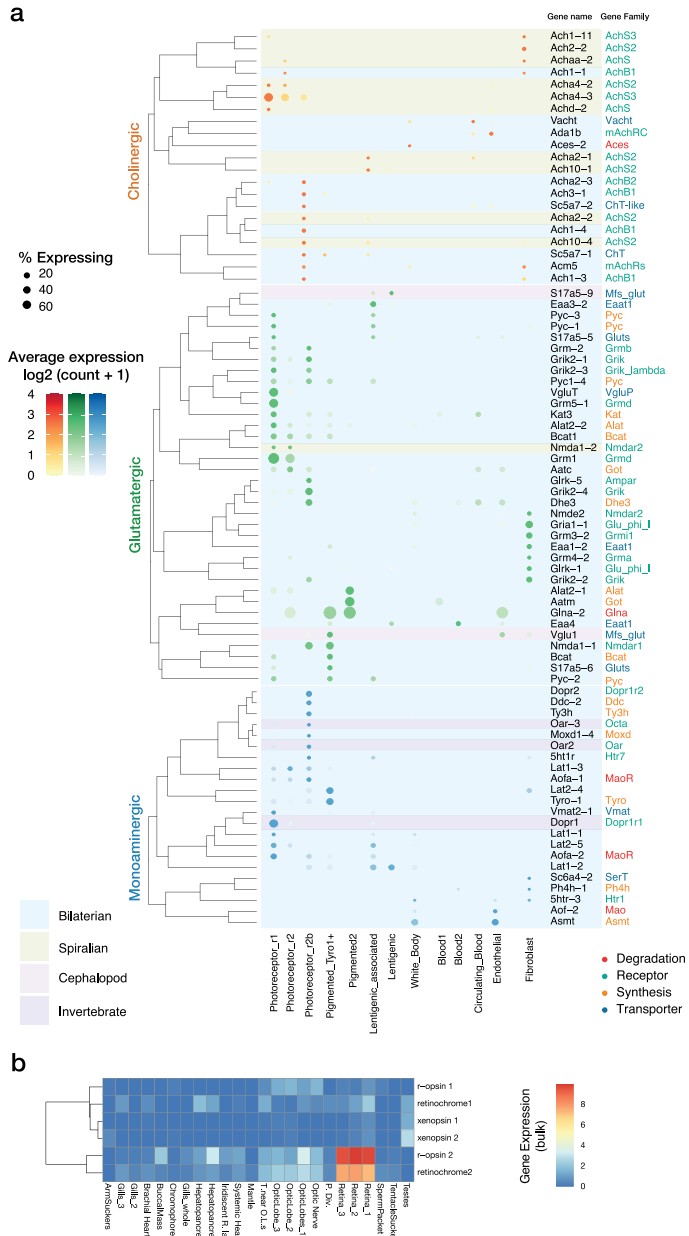

**Extended Data Fig. 3 | Gene expression in *E. berryi* retina. a**, Dot plot of neuronal markers (in retina scRNA-seq dataset). Circle radius is proportional to fraction of cells in each cluster expressing gene, and colour intensity is specified by average scaled expression level as noted. **b**, Heatmap of log$_{10}$(FPKM + 1) bulk RNA-seq for 6 chromoproteins r-opsin1, r-opsin2, retinochrome-1, retinochrome-2, xenopsin1, xenopsin2 in different organs of a male adult *E. berryi*.

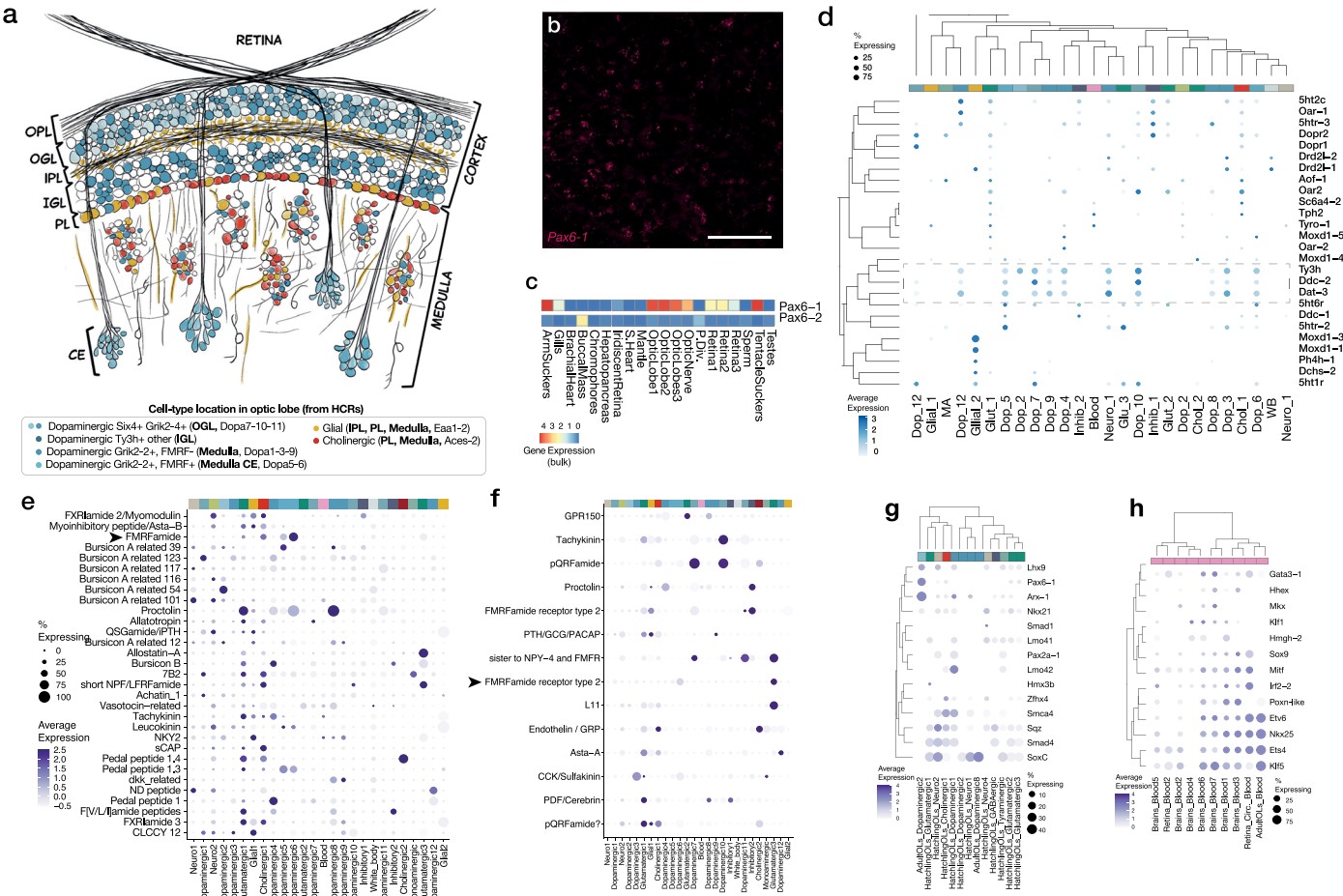

**Extended Data Fig. 4 | Gene expression in *E. berryi* optic lobe. a**, Schematic optic lobe neuroanatomy and summary of cell localisation from HCR (see Fig. 3). Abbreviations: OPL - outer plexiform layer, OGL - outer granule layer, IPL - inner plexiform layer, IGL - inner granule layer, PL - palisade layer, CE - centrifugal. **b-c**, Expression of Pax6 orthologues in *E.berryi* on selected organs. **b**, HCR staining for Pax6-1 RNA on sections of optic lobe from 10dph individuals (reproduced n > 3). Scale bar 50 µm. **c**, heatmap showing expression levels of both Pax6-1 and Pax6-2 orthologues in bulk RNA in $\log_2(\text{FPKM} + 1)$. **d**, Dot plot indicating expression

levels of monoamine neurotransmission genes in the mature adult optic lobes of *E. berryi*. Box with dotted lines surrounds expression of genes involved in synthesis of neurotransmitters. **d**, Dotplot indicating expression levels of transcription factors in adult and hatchling optic lobe datasets. **e-f**, Dotplots showing expression levels of neuropeptides ligands **e**, and receptors **f**, in each optic lobe cell cluster. **g-h**, Dotplots showing expression of selected transcription factors in populations of hatchling and mature optic lobes marking neuronal cell populations **g**, and hemocyte cell populations, **h**.

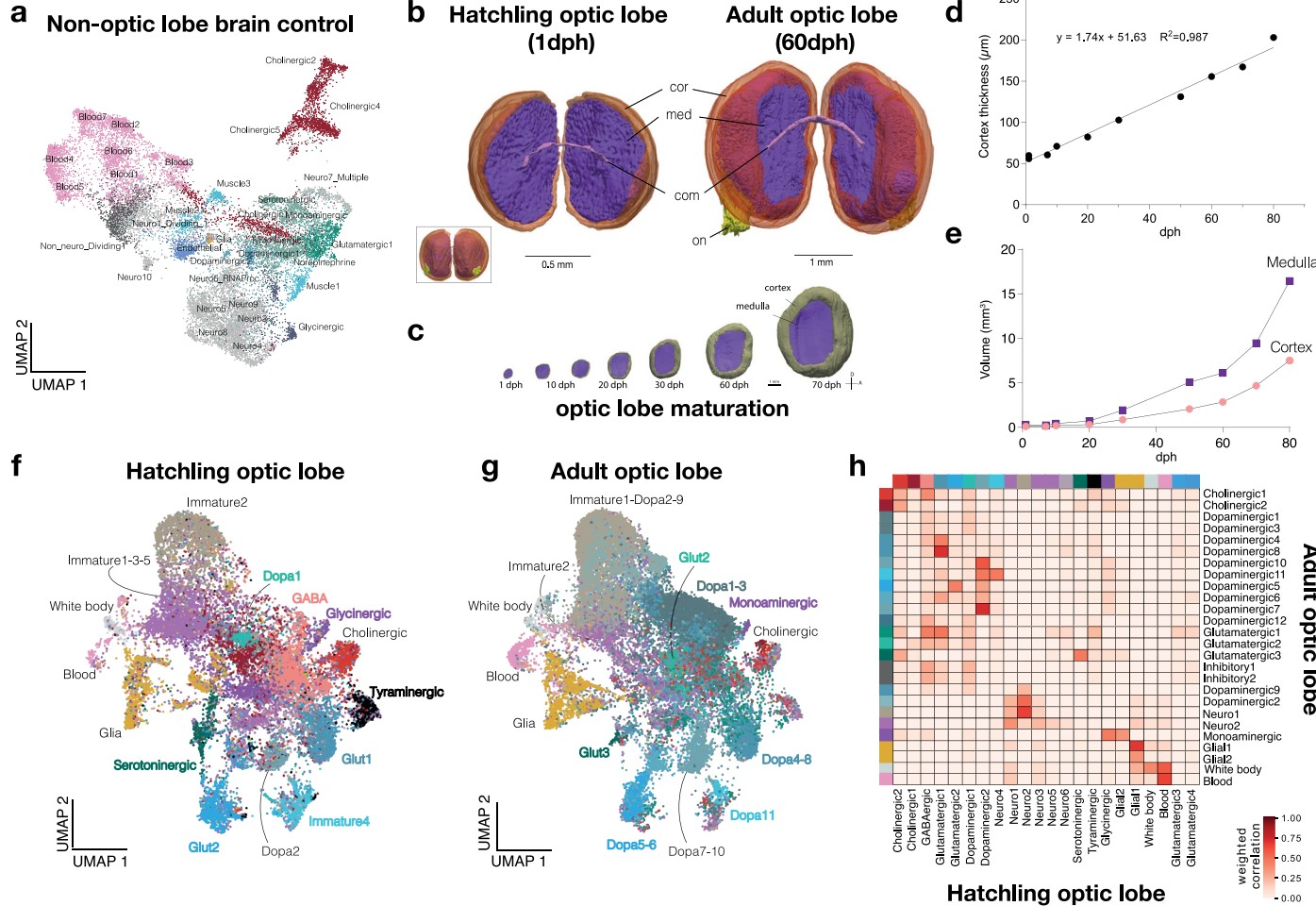

**Extended Data Fig. 5 | Optic lobe maturation and change in neurotransmitter usage. a**, UMAP of 17,468 cells *E. berryi* non-optic lobe perioesophageal brain control dataset. **b-e**, Evidence of optic lobe maturation and growth based on microCT imaging. **b**, Comparison of optic lobe morphology by microCT segmentation and rendering of optic lobes at 1 day post hatching (left) and 60 days post hatching (right) - posterior view. Cortex is colored in pink with transparency while medulla that lies beneath it is shown in purple (red colored surface corresponds to cortex wrapping the medulla): the bending of the optic lobe causes the cortex to engulf a larger fraction of the optic lobe. Inlet shows anterior view of optic lobes at 1 dph with minute optic nerve located ventrally (yellow). **c**, 3D rendering of relative cortex (beige) and medulla (purple) growth

over time from 1 dph to 70 dph. **d-e**, Plot of measurements made based on microCT scans from individuals aged 1 dph to 80 dph of the thickness of the cortex in µm **d**, and of the volume of both the cortex and the medulla in mm³. **f-g**, UMAP projection of integrated **f**, hatchling and **g**, adult optic lobe single-cell data. Note that clustering was performed separately (see Figs. 3a and 4a). Coloured cluster labels highlight cell clusters with distinct neurotransmitter usage in adults as opposed to hatchlings. **h**, Gene expression similarity across clusters of the adult and hatchling optic lobe datasets, using weighted Pearson correlation. Cell cluster colours at the left (adult) and top (hatchling) of the heatmap match the colours used in the UMAP.

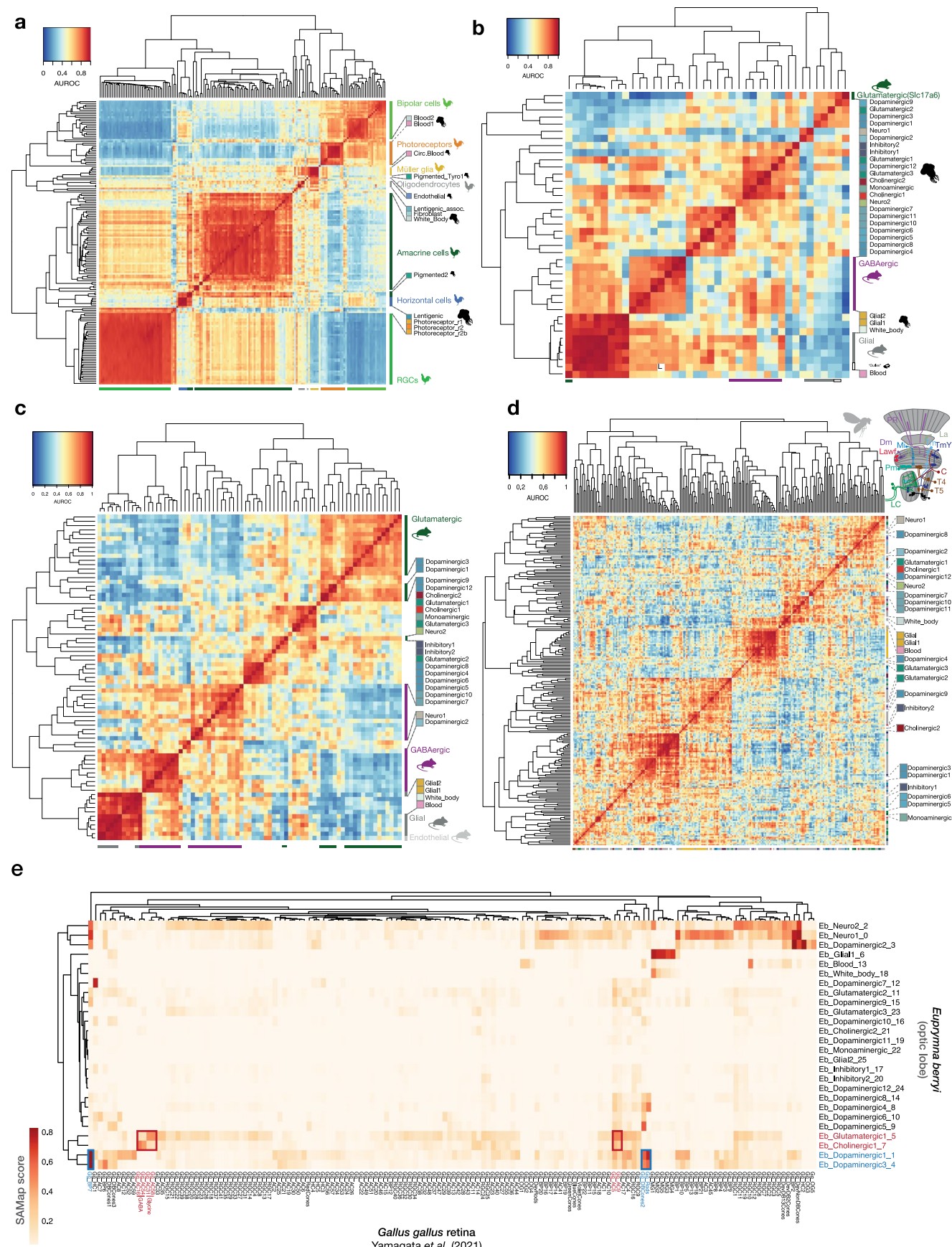

**Extended Data Fig. 6 | See next page for caption.**

**Extended Data Fig. 6 | Comparisons of *E. berryi* visual system single-cell transcriptomes with vertebrate retina and visual cortex single-cell transcriptomes. a-d**, Heatmaps displaying MetaNeighbor AUROC scores or "Reciprocal_top_hit" match types are identified as cell type pairs comparisons between **a**, *E. berryi* retina and chicken retina cell types (Yamagata et al., 2021)

**b-d**, *E. berryi* adult optic lobe comparisons to b, mouse dorsal lateral geniculate nucleus (Allen Brain Atlas), **c**, mouse visual cortex[129] and **d**, adult *Drosophila* optic lobes[75]. **e**, SAMap[78] comparison of *E. berryi* adult optic lobe and chicken retina cell types[74]. Silhouettes from PhyloPic under a Creative Commons license CC0 1.0: chicken, Steven Traver; mouse, Jiro Wada; *Drosophila*, Ramiro Morales-Hojas.

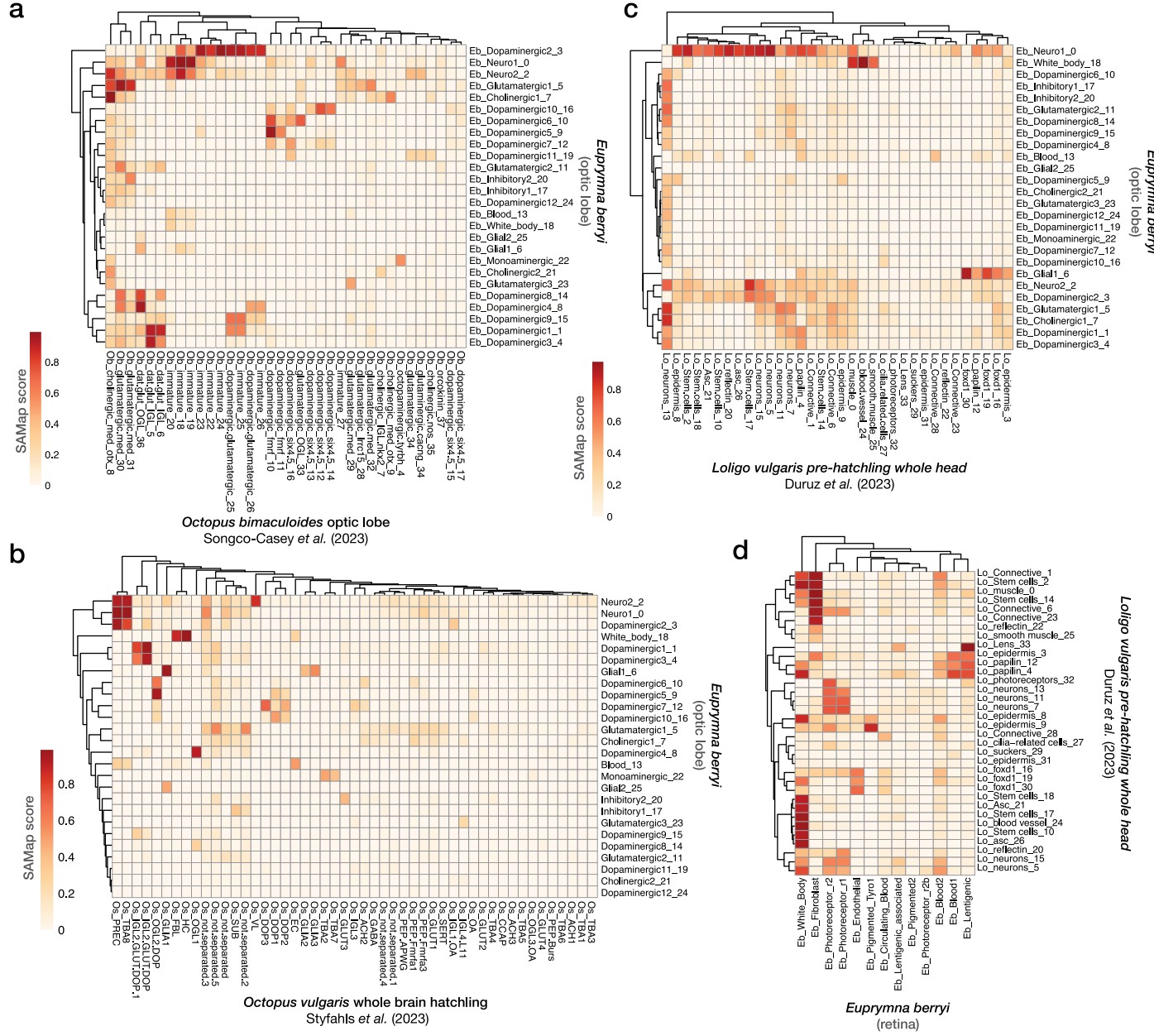

**Extended Data Fig. 7 | Comparisons of E. berryi visual system single-cell transcriptomes with single-cell transcriptomes from other cephalopods. a-d,** Heatmaps showing SAMap[78] single-cell expression comparisons between cell populations of *E. berryi* adult optic lobe and a, *O. bimaculoides* optic lobe[99], **b**, *O. vulgaris* hatchling whole brain[100], **c**, *L. vulgaris* pre-hatchling whole head[101], **d**, between *E. berryi* adult retina and *L. vulgaris* pre-hatchling whole head[101].

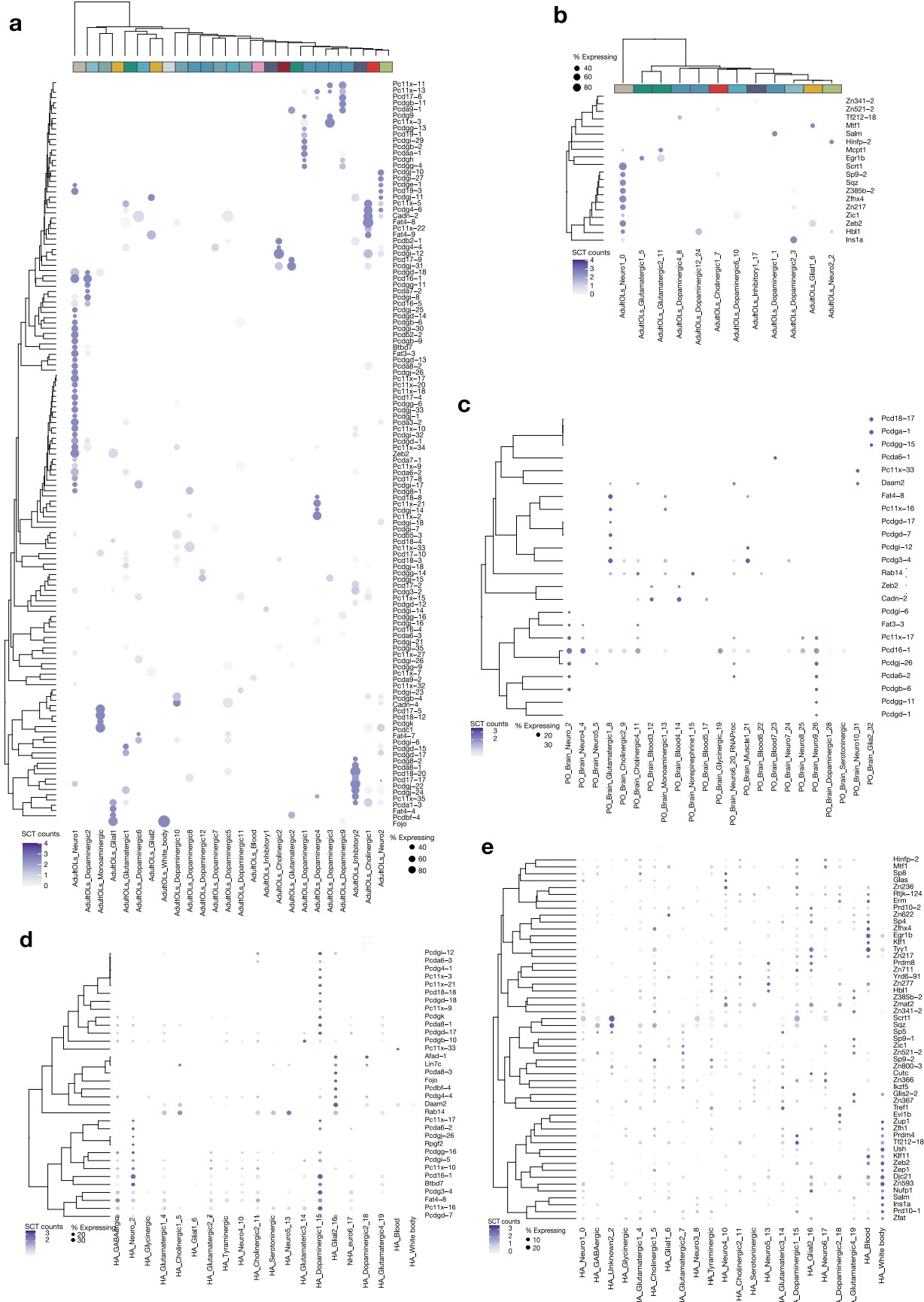

**Extended Data Fig. 8 | Cell type complexity examined through protocadherin and transcription factor expression. a**, Dotplot showing expression levels of protocadherin molecules in mature adult optic lobes **b**, Dotplot showing C2H2 transcription factor expression in mature adult optic lobe. **c-d**, Dotplots showing expression of protocadherins in **c**, perioesophageal brain and **d**, hatchlings optic lobes. **e**, Dotplot of C2H2 transcription factor expression in hatchling optic lobes.

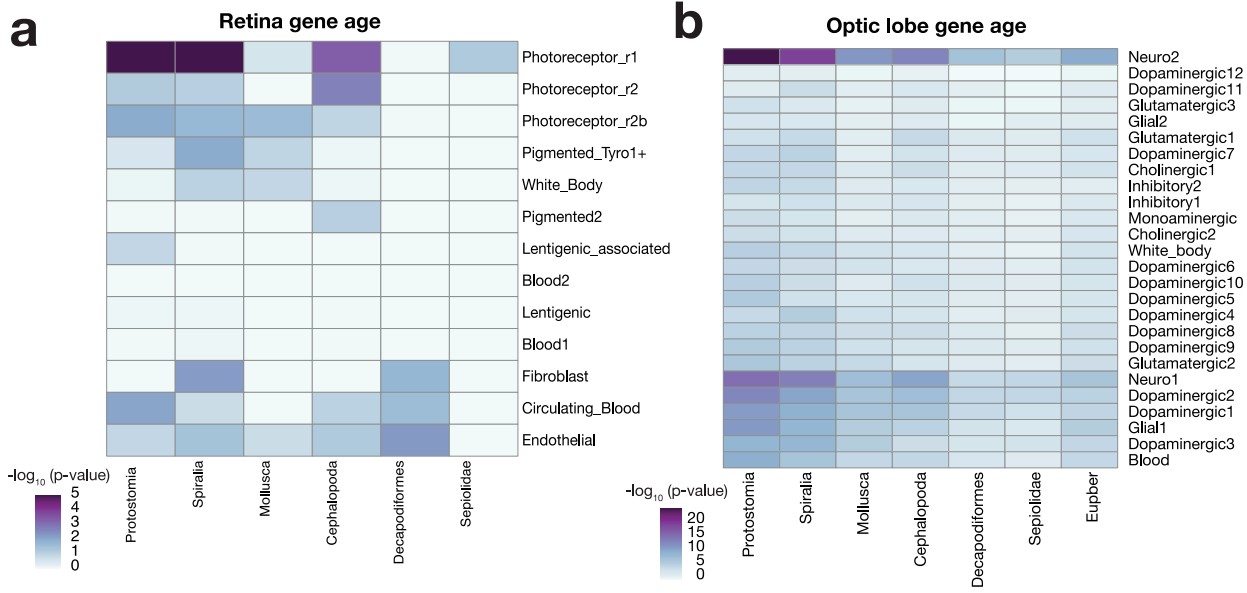

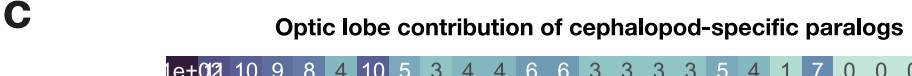

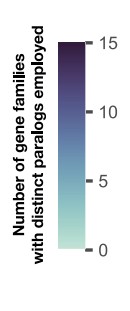

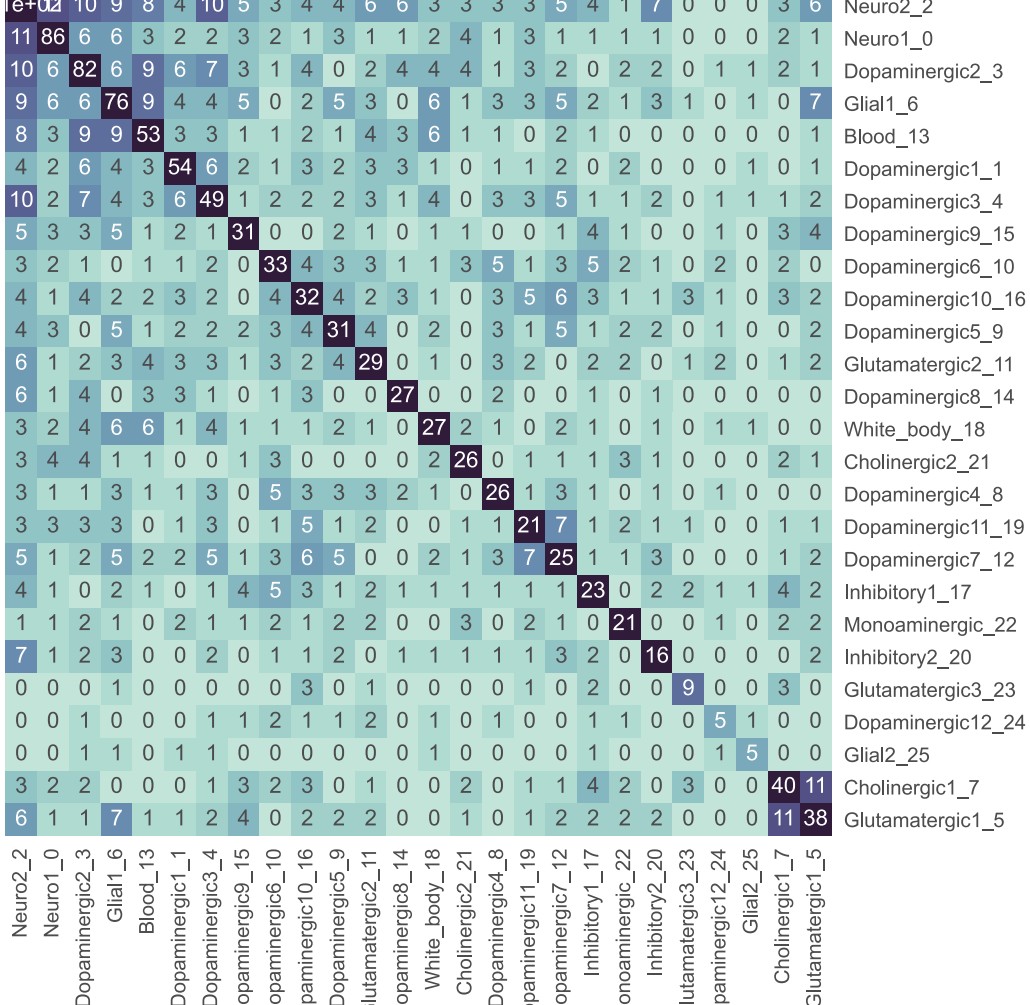

**Extended Data Fig. 9 | Evolutionary history of genes employed by the *E. berryi* visual system. a-b**, Heatmaps showing -log10(p-value) of enrichment test (2-sides Fisher exact test) of gene family origin of marker genes for retinal **a**, and adult optic lobe **b**, cell populations. **c**, Contribution of cephalopod paralogs to *E.* *berryi* optic lobe cell type diversification. Numbers in the heatmap indicate the number of cephalopod paralogous gene families for which distinct paralogs are specifically expressed in distinct cell populations.

| | |
|---|---|

# Reporting Summary

## Statistics

For all statistical analyses, confirm that the following items are present in the figure legend, table legend, main text, or Methods section.

| n/a | Confirmed | |
|---|---|---|
| ☐ | ☒ | The exact sample size (*n*) for each experimental group/condition, given as a discrete number and unit of measurement |
| ☐ | ☒ | A statement on whether measurements were taken from distinct samples or whether the same sample was measured repeatedly |
| ☐ | ☒ | The statistical test(s) used AND whether they are one- or two-sided *Only common tests should be described solely by name; describe more complex techniques in the Methods section.* |
| ☒ | ☐ | A description of all covariates tested |
| ☐ | ☒ | A description of any assumptions or corrections, such as tests of normality and adjustment for multiple comparisons |
| ☐ | ☒ | A full description of the statistical parameters including central tendency (e.g. means) or other basic estimates (e.g. regression coefficient) AND variation (e.g. standard deviation) or associated estimates of uncertainty (e.g. confidence intervals) |
| ☐ | ☒ | For null hypothesis testing, the test statistic (e.g. *F*, *t*, *r*) with confidence intervals, effect sizes, degrees of freedom and *P* value noted *Give P values as exact values whenever suitable.* |
| ☒ | ☐ | For Bayesian analysis, information on the choice of priors and Markov chain Monte Carlo settings |
| ☒ | ☐ | For hierarchical and complex designs, identification of the appropriate level for tests and full reporting of outcomes |
| ☐ | ☒ | Estimates of effect sizes (e.g. Cohen's *d*, Pearson's *r*), indicating how they were calculated |

*Our web collection on statistics for biologists contains articles on many of the points above.*

## Software and code

Policy information about availability of computer code

| Data collection | SRA-Toolkit v2.11.3 |
|---|---|
| Data analysis | Jellyfish v.2.2.7,  bwa mem v.0.7.17, wtdbg2 v.2.5, Racon v.1.3.2, Scaff10X, Merqury v.1.1, minimap2 v.2.16, IsoSeq-3.0, BUSCO v.3.1.0, HiRise, Juicer v.e0d1bb7, Juicebox v.1.11.08, Star v.2.5.2,bcl2fastq (v2.19),  Stringtie v1.3.3b,  GMAP version 2019-02-26, Mikado v1.2.1, Taco, Portcullis v1.0.2, TransDecoder, Augustus v.3.3.3, RepeatModeler v.1.0.11, RepeatMasker v.4.0.7, PASA v.2.41, PfamScan v.1.6, Orthofinder v2.33,cellranger mkfastq v.3.1.0, UTR_extension_GTF.py v.1, R version 4.1.2, Seurat v.4.1.0, Amira v.6.5, ParaView , Fiji v.1.2.30/1.53, SAMtools v.1.9, BEDtools v2.25.0, Python 3.8.10,FigTree v.1.4.4,RAxML v.8.2.11.9, Adobe Photoshop 2021 22.4.3. release, Adobe Illustrator 2021 25.4.1 release |

For manuscripts utilizing custom algorithms or software that are central to the research but not yet described in published literature, software must be made available to editors and reviewers. We strongly encourage code deposition in a community repository (e.g. GitHub). See the Nature Portfolio guidelines for submitting code & software for further information.

## Data

Policy information about availability of data

All manuscripts must include a data availability statement. This statement should provide the following information, where applicable:

- Accession codes, unique identifiers, or web links for publicly available datasets
- A description of any restrictions on data availability
- For clinical datasets or third party data, please ensure that the statement adheres to our policy

Accession code and unique identifiers to previously publicly available datasets are listed in Methods.  All sequence data associated with this project are available at the European Nucleotide Archive (project  PRJEB52690) and Gene Expression Omnibus (accession number GSE203527).

## Research involving human participants, their data, or biological material

Policy information about studies with human participants or human data. See also policy information about sex, gender (identity/presentation), and sexual orientation and race, ethnicity and racism.

| | |
|---|---|
| Reporting on sex and gender | Use the terms sex (biological attribute) and gender (shaped by social and cultural circumstances) carefully in order to avoid confusing both terms. Indicate if findings apply to only one sex or gender; describe whether sex and gender were considered in study design; whether sex and/or gender was determined based on self-reporting or assigned and methods used. Provide in the source data disaggregated sex and gender data, where this information has been collected, and if consent has been obtained for sharing of individual-level data; provide overall numbers in this Reporting Summary.  Please state if this information has not been collected. Report sex- and gender-based analyses where performed, justify reasons for lack of sex- and gender-based analysis. |
| Reporting on race, ethnicity, or other socially relevant groupings | Please specify the socially constructed or socially relevant categorization variable(s) used in your manuscript and explain why they were used. Please note that such variables should not be used as proxies for other socially constructed/relevant variables (for example, race or ethnicity should not be used as a proxy for socioeconomic status). Provide clear definitions of the relevant terms used, how they were provided (by the participants/respondents, the researchers, or third parties), and the method(s) used to classify people into the different categories (e.g. self-report, census or administrative data, social media data, etc.) Please provide details about how you controlled for confounding variables in your analyses. |
| Population characteristics | Describe the covariate-relevant population characteristics of the human research participants (e.g. age, genotypic information, past and current diagnosis and treatment categories). If you filled out the behavioural & social sciences study design questions and have nothing to add here, write "See above." |
| Recruitment | Describe how participants were recruited. Outline any potential self-selection bias or other biases that may be present and how these are likely to impact results. |
| Ethics oversight | Identify the organization(s) that approved the study protocol. |

Note that full information on the approval of the study protocol must also be provided in the manuscript.

# Field-specific reporting

Please select the one below that is the best fit for your research. If you are not sure, read the appropriate sections before making your selection.

☒ Life sciences          ☐ Behavioural & social sciences          ☐ Ecological, evolutionary & environmental sciences

For a reference copy of the document with all sections, see nature.com/documents/nr-reporting-summary-flat.pdf

# Life sciences study design

All studies must disclose on these points even when the disclosure is negative.

| | |
|---|---|
| Sample size | Tissues were sampled and dissociations were carried out using tissues from a single individual at a time. |
| Data exclusions | No data was excluded. |
| Replication | scRNA-seq was performed in replicates (4, 5, 6 , 5 for mature OLs, hatchling OLs, retinas, non-optic lobe perioesophageal brain respectively) |
| Randomization | All animal collections were performed randomly to ensure genetic variability. |
| Blinding | All animal collections were allocated blindly to any of the replicates of study. |

# Reporting for specific materials, systems and methods

We require information from authors about some types of materials, experimental systems and methods used in many studies. Here, indicate whether each material, system or method listed is relevant to your study. If you are not sure if a list item applies to your research, read the appropriate section before selecting a response.

## Materials & experimental systems

| n/a | Involved in the study |
|-----|----------------------|
| ☐ | ☒ Antibodies |
| ☒ | ☐ Eukaryotic cell lines |
| ☒ | ☐ Palaeontology and archaeology |
| ☐ | ☒ Animals and other organisms |
| ☒ | ☐ Clinical data |
| ☒ | ☐ Dual use research of concern |
| ☒ | ☐ Plants |

## Methods

| n/a | Involved in the study |
|-----|----------------------|
| ☒ | ☐ ChIP-seq |
| ☒ | ☐ Flow cytometry |
| ☒ | ☐ MRI-based neuroimaging |

## Antibodies

| | |
|---|---|
| Antibodies used | Rb Anti-FMRF-amide (Ab15348) 1:1000; Alexa 568 goat anti-rabbit IgG 1:250; Anti-acetylated tubulin (ab24610) 1:100; Alexa 568 goat anti-mouse (1:250) |
| Validation | Antibody cross-reactivity against Eberryi was predicted based on multiple sequence alignments (MSA) of targeted antigens with closely phylogenetically related species.Antibodies were then validated in immunohistochemistry with our own species. |

## Animals and other research organisms

Policy information about studies involving animals; ARRIVE guidelines recommended for reporting animal research, and Sex and Gender in Research

| | |
|---|---|
| Laboratory animals | Euprymna berryi. We kept animals caught in the wild from both sexes and and all ages. Female individuals carrying eggs were permitted to lay eggs and individuals were reared to adulthood. |
| Wild animals | Adult Euprymna berryi of unknown ages were collected from the coast of Mie prefecture in Japan and transported to the lab. Animals were euthanized in the course of study using 4% ethanol in sterile-filtered natural seawater. |
| Reporting on sex | Genome was obtained utilising sperm from male individual. Due to unequal male:female ratio, male specimens were utilised for scRNA-seq, bulk RNA-seq, to preserve the few female individuals in culture. Sex of individuals at embryonic stages cannot be determined as they lack hectocotylus at this stage and is therefore unknown. |
| Field-collected samples | Adult Euprymna berryi of unknown age were collected from the coast of Mie prefecture in Japan and transported to Okinawa where they were acclimated to temperature (20°C) and pH (8.3) of a closed aquarium system in filtered natural seawater obtained from the shores of Okinawa, Japan (OIST Seragaki Marine Science Station). Animals were maintained essentially as described previously until they were sacrificed for experiments. Animals were exposed to a static 12:12 hour light:dark cycle. Tanks that housed the animals contained an enriched environment including natural substrate (autoclaved sand or crushed coral), parts of clay pots and natural rocks as dens. Mature animals were fed daily with Opossum shrimp or mysids, whereas Neomysis Japonica proved a suitable prey for hatchlings. Fresh glass shrimp, Palaemonetes spp. and frozen shrimp which were purchased in local grocery stores were fed to late juveniles and adults. Tanks were cleaned daily to remove uneaten food and waste matter. Prior to experiments, animals were euthanized using 4% ethanol in sterile-filtered natural seawater. Animals were allowed to breed freely. Hatchlings were obtained either from eggs provided by females impregnated in the wild or by breeding wild animals in the laboratory. |
| Ethics oversight | This study was carried out in accordance with procedures authorised by Guidelines for Proper Conduct of Animal Experiments by the Science Council of Japan. Despite the absence of legislation pertaining specifically to cephalopods in Japan, we aspired to abide by the highest standards in the field. All conducted experiments were therefore also in line with EU Directive 2010/63/EU and with the guidelines and the principles detailed in Anrews et al., 2013, Smith et al., 2013, Fiorit et al., 2015 and Dicristina et al., 2015. All experiments were approved by the Okinawa Institute of Science and Technology Graduate University Animal Care and Use Committee (approval ID: 2018-204). No transgenic animals were used in this study. |

Note that full information on the approval of the study protocol must also be provided in the manuscript.

# Plants

| | |
|---|---|
| Seed stocks | *Report on the source of all seed stocks or other plant material used. If applicable, state the seed stock centre and catalogue number. If plant specimens were collected from the field, describe the collection location, date and sampling procedures.* |
| Novel plant genotypes | *Describe the methods by which all novel plant genotypes were produced. This includes those generated by transgenic approaches, gene editing, chemical/radiation-based mutagenesis and hybridization. For transgenic lines, describe the transformation method, the number of independent lines analyzed and the generation upon which experiments were performed. For gene-edited lines, describe the editor used, the endogenous sequence targeted for editing, the targeting guide RNA sequence (if applicable) and how the editor was applied.* |
| Authentication | *Describe any authentication procedures for each seed stock used or novel genotype generated. Describe any experiments used to assess the effect of a mutation and, where applicable, how potential secondary effects (e.g. second site T-DNA insertions, mosiacism, off-target gene editing) were examined.* |

