## [Peer Review File · Nature Ecology & Evolution]

A single cell atlas of bobtail squid visual and nervous system highlights molecular principles of convergent evolution

Corresponding Author: Dr Daria Gavriouchkina

Version 0:

Decision Letter:

23rd September 2022

Dear Daria,

Your manuscript entitled "A single-cell atlas of bobtail squid visual and nervous system highlights molecular principles of convergent evolution" has now been seen by 3 reviewers, whose comments are attached. The reviewers have raised a number of concerns which will need to be addressed before we can offer publication in Nature Ecology & Evolution. We will therefore need to see your responses to the criticisms raised and to some editorial concerns, along with a revised manuscript, before we can reach a final decision regarding publication.

We therefore invite you to revise your manuscript taking into account all reviewer and editor comments. Please highlight all changes in the manuscript text file in Microsoft Word format.

* If you have not done so already please begin to revise your manuscript so that it conforms to our Article format instructions at <http://www.nature.com/natecolevol/info/final-submission>. Refer also to any guidelines provided in this letter.

Link Redacted

Nature Ecology & Evolution is committed to improving transparency in authorship. As part of our efforts in this direction, we are now requesting that all authors identified as 'corresponding author' on published papers create and link their Open Researcher and Contributor Identifier (ORCID) with their account on the Manuscript Tracking System (MTS), prior to acceptance. ORCID helps the scientific community achieve unambiguous attribution of all scholarly contributions. You can create and link your ORCID from the home page of the MTS by clicking on 'Modify my Springer Nature account'. For more

information please visit please visit www.springernature.com/orcid.

[redacted]

Reviewer expertise:

Reviewer #1: cephalopod nervous system, genomics

Reviewer #2: evolution of cell types, genomics, scRNAseq

Reviewer #3: evolution of the nervous system

Reviewers' comments:

Reviewer #1 (Remarks to the Author):

The report by Gavriouchkina and colleagues is expertly done. It includes the first report of a high-quality genome for *Eupryma berryi*, arguably the highest quality cephalopod genome yet released. The single-cell datasets for retina and optic lobe are analyzed to a particularly high level. Unfortunately, particularly given that a chromosomal level assembly is published for the sister species *E. scolopes*, this submission represents more of a research resource report than a set of important findings in the field of evolution.

The authors aim to build on the proposal by Cajal, followed by Young, that the cephalopod optic lobe (OL) should be viewed as equivalent to the "deep retina" of the vertebrate eye. It is of little doubt that the OL is, in the sense of being a secondary processing center, like the inner vertebrate retina, but whether similar (convergent) processing is going on is unknown and not aided by this single-cell data, at least not so far. The suggestion that one molecularly defined cell type may correspond to Young's bipolar cells (which may not do anything like what the vertebrate bipolar cell does) or that the long-known centrifugal projection to the retina could use FMRFamide as a messenger, are interesting predictions, but are neither confirmed nor do they aid the argument for convergence. There are many other conclusions, such as that their retina data confirm the relative simplicity of the cephalopod retina, but these are well-known, at least in the cephalopod field.

This manuscript, available earlier this year on bioRxiv, joins other bioRxiv single-cell reports on cephalopods from the Sprecher, Niell and Seuntjens groups. These papers are cited but only just. In particular, the Seuntjens paper has a lot to say about octopus glial cells, and the Niell paper covers cell-type identity and development in the octopus OL. It might be profitable for the report under review to move beyond simple citation and highlight for the field what in their analysis aligns with the octopus findings, and what departs from it.

Reviewer #2 (Remarks to the Author):

Gavriouchkina and co-authors introduce a single-cell transcriptomic atlas of the bobtail squid visual and nervous systems. They combine this extensive single-cell sampling with a high-quality genome assembly and annotation (an outstanding resource), as well as extensive imaging and HCR validations. In addition to the high quality of the data, I really appreciate the way the results are carefully contextualized and interpreted.

Besides the interest that this study has for understanding cephalopod biology, the studied system represents an interesting case-example of phenotypic convergence and the authors take this opportunity to explore the molecular signatures underlying this process. Unfortunately, in my opinion this comparative aspect of the manuscript is lacking in details and would benefit from a more thorough and quantitative analysis. Overall, though, I think this an excellent and interesting study and a valuable resource. Therefore, I find it suitable for publication in NEE.

These are my main criticisms/comments, which I hope can help improve the clarity and consistency of the paper:

- Have the authors tried to jointly cluster all datasets? It should be a more elegant way of presenting the results (e.g. sorting out blood cells together from the retina and control blood experiments) and then zooming into interesting subpopulations as needed – as it is commonly done in single-cell studies.
- Related to the previous point: how batchy is the single-cell data from different preparations and 10x runs?
- I wonder if there are any genes exclusively expressed in the "photoreceptor_r2" population. In contrast to this, and looking at Figure S3, I find it very strange that the tiny "photoreceptor_undiff" population expresses so very many specific markers, how do the authors explain an undifferentiated/progenitor population with so many and so specific effector genes?
- Continuing with the retina dataset, any idea what is the "contractile" population? What makes the authors think this is contractile?
- The analysis of adult vs 1dph hatchling datasets (Figure 4) is rather limited. The authors should either analyze them

together or show a pairwise comparison (e.g. using gene expression correlations or jaccard intersections of shared markers) between these two datasets.

- The cross-species comparisons would benefit from a less descriptive/more quantitative approach. For example, identifying how many and which specific gene orthologs (or paralogs of the same homology group) are shared between clustered chicken-squid cell types, including TFs. We never get to see the co-expression of markers described by the authors. One example of how the paper would benefit from this is the “bipolar-cell gene expression module” that the authors describe in the discussion – can you retrieve the genes that would compose this module from your cross-species analysis?
- Similarly, the paper is missing a quantitative perspective on the “subfunctionalization” question, which remains highly speculative/descriptive. I would recommend omitting this bit or providing a dedicated quantitative analysis and visualization to support this (otherwise interesting) point.
- I would like to see more details in the methods, particularly regarding gene orthology and cross-species comparison (e.g. the motivation behind using Metaneighbors and the consistency of alternative approaches).
- The authors MUST unify the way gene expression (both single-marker plots and multi-marker maps) across cell types is visualized, e.g. cluster-level normalized expression in Figure S2e-f; % expressing cells dotplots in Figure S3; violin plots in Figure 3; and single-cell-level normalized expression in Figure 1. Figure S4 is particularly chaotic example of this. Choose one style and stick to it.
- Please, make sure all figure legends and axes are labeled (e.g. Figure S2, y-axis in Figure 3b).
- Small typo: Line 939: “protomes” -> “proteomes”

Reviewer #3 (Remarks to the Author):

This paper introduces the genome and single cell transcriptomic analysis of the cephalopod, *Euprymna berryi*. The authors carried out single cell sequencing for cells of the retina, optic lobe and non-neural tissues for adult and juvenile animals. Neuronal and other cells were identified by canonical markers, and neurons were subdivided based on transmitter expression/reception. The work establishes a rich and valuable resource for developmental-genetic studies of cephalopod neural development. The authors also use the data to confirm previous findings, in particular the cell typing of the retina, the idea of considering the optic lobe as a “deep retina”, and peptidergic feed back innervation of the retina by the optic lobe. The paper adds significant new insights into the development and evolution of the bilaterian visual system. Findings are generally well described and documented. My comments:

1. In the introduction and throughout the Results sections the authors need to provide more context when presenting details of cephalopod neuroanatomy.

1.1. Introduce apparent simplicity in terms of photoreceptor types; given the large cephalopod retina (millions of photoreceptors), why are there only 1-2 receptor types? How is color vision achieved? Have previous works commented on this paucity of photoreceptors (as well as support cells)?

1.2. The support cells appear to produce pigment. What about glial cells in the retina?

1.3. Summarize existing structurally/transmitter defined neuron types of the optic lobe.

A schematic showing the organization of the optic lobe, with layering (cell bodies vs neuropil) and structurally defined cell types, similar to the schematic of the retina in Fig.2h, would be very important. How is the medulla organized? It does not seem layered (Fig.3); cell body clusters appear to be interspersed with domains of neuropil.

1.4. Mainly transmitters are used to differentiate neuron classes. Has the expression/localization of these transmitters been studied based on (immune)histochemistry in cephalopods in previous studies? Have these transmitters been assigned to morphological cell classes? If so, do the current localization findings confirm or contradict existing studies?

1.5. Providing more background on neuroanatomy is particularly important in regard to interconnections between photoreceptors and optic lobe neurons. In the vertebrate (or arthropods) there is a strict layering, where photoreceptors project on to bipolar cells, which in turn target the output neurons. The authors cite previous neuroanatomical findings that “bipolar cells” were identified in cephalopod optic lobes. These findings need to be summarized/presented schematically (see above). Does the localization of putative molecular markers for bipolar cells match the descriptions by Young?

2. Glia: Glial cell types have been characterized in many bilaterians by morphology and interaction with other cell types; examples are the astrocytes/astrocyte-like cells in vertebrates and insects (interacting with neurites and synapses and synapses in the neuropil), oligodendroglia (myelin sheaths) in vertebrates, cortex glia (ensheathing cell bodies) in insects, etc. Please provide an overview/schematic of existing studies (EM analysis and others) that characterize glia in cephalopods

3. Evolution:

3.1. The authors place their findings in the framework of the “deep retina” hypothesis, whereby vertebrate retinal neuronal layers are “displaced” into the optic lobe. This idea has been discussed many times also in regard to arthropod eyes in which, like in cephalopod eyes, the retina is formed only by photoreceptors, and no interneurons. An inference from the deep retina hypothesis would be that during vertebrate evolution, brain neurons (e.g., interneurons targeted by photoreceptors) become “outsourced” into the retina. An alternative view has been presented (e.g., Arendt: Evolution of eyes and photoreceptor cell types. *Int J Dev Biol* 2003) that postulates that at the onset of eye evolution, photoreceptors carried out all visual function; receptors then diversified into different types, as well as neurons. Gene expression data were also used to defend this “shallow retina” hypothesis. Given the scope and importance of the current paper for the cephalopod visual system and visual system evolution in general, this alternate hypothesis needs to be exposed in the introduction. Are there data that would more clearly help (or defeat) the shallow retina hypothesis?

3.2. For example, in line 448, the authors mention a similarity in gene expression between vertebrate retinal ganglion cells and squid photoreceptors: does this not support the shallow retina hypothesis (photoreceptors diversifying into neurons that form deep layers of retina)?

*****END*****

Version 1:

Decision Letter:

13th February 2025

Dear Daria,

Thank you for submitting your revised manuscript "A single-cell atlas of bobtail squid visual and nervous system highlights molecular principles of convergent evolution" (NATECOLEVOL-220716960A). It has now been seen again by two of the original reviewers and their comments are below. The reviewers find that the paper has improved in revision, and therefore we'll be happy in principle to publish it in Nature Ecology & Evolution, pending minor revisions to comply with our editorial and formatting guidelines.

[redacted]

Reviewer #2 (Remarks to the Author):

The authors have thoroughly addressed all my comments and suggestions. Congratulations on an excellent study.

Reviewer #3 (Remarks to the Author):

The authors have done a remarkable job in addressing my previous comments, adding text and figure panels that provide more context and discussion of previous data. I find the work suitable for publication.

Reviewer #1 (Remarks to the Author):

The report by Gavriouchkina and colleagues is expertly done. It includes the first report of a high-quality genome for *Euprymna berryi*, arguably the highest quality cephalopod genome yet released. The single-cell datasets for retina and optic lobe are analyzed to a particularly high level. Unfortunately, particularly given that a chromosomal level assembly is published for the sister species *E. scolopes*, this submission represents more of a research resource report than a set of important findings in the field of evolution.

We would like to thank Reviewer #1 for the positive comments about the superior quality of our genome sequence and the high level of our single-cell analysis of the bobtail squid retina and optic lobe. While our manuscript does report these resources, our main goal is to present important findings related to the evolution of complex visual systems based on comparative analyses of these data. We note that this aim is clearly distinct from studies published on chromosome evolution in *E. scolopes*.

More specifically, our work characterises the cell types of the retina and optic lobe of the bobtail squid and analyses their convergence with the vertebrate retina. While there is logically no necessary 1:1 correspondence between the cell types of these two deeply divergent systems, we find that some cell types of the optic lobe (Dopaminergic1-3) share global expression similarity with the bipolar neurons of the vertebrate retina, while other share similarity with retinal ganglion cells (Dopaminergic4,5,6,7,8,10,11). This observation provides support for the scenario proposing that the ancestral visual system in bilaterian animals featured two dedicated neuronal cell populations.

We also find a broad diversity of cell types, including multiple dopaminergic neurons emerging independently in the bobtail squid optic lobe. These dopaminergic neurons are involved in retrograde signalling to control the photoreceptors in the retina. The tight integration of optic lobe and retina corroborates Ramón y Cajal's deep retina hypothesis. Curiously, we observed that, although there is a surprising diversity of neurons in the cephalopod visual system, the squid, vertebrate and fly visual systems express the same broad classes of genes. We therefore further investigated the mechanisms that rendered this diversity possible.

We find that conserved genes are co-expressed in specific neurons in the same pattern only in the broadest sense. For example, cells can be identified by the neurotransmitter(s) they produce and receive based on batteries of co-expressed genes involved in neurotransmitter biosynthesis/transport/degradation and receptors, respectively. Yet the cell types themselves do not appear to be in a simple correspondence, and different gene duplications give rise to divergent functions and cell types. These findings show that (contrary to some published claims about deep cellular homology in other systems) there has been considerable cell type turnover and "invention" in vertebrates, flies, and cephalopods. We believe that this is an important conceptual point that will influence future studies of cell-type evolution.

The authors aim to build on the proposal by Cajal, followed by Young, that the cephalopod optic lobe (OL) should be viewed as equivalent to the "deep retina" of the vertebrate eye. It is

of little doubt that the OL is, in the sense of being a secondary processing center, like the inner vertebrate retina, but whether similar (convergent) processing is going on is unknown and not aided by this single-cell data, at least not so far.

We thank Reviewer #1 for raising an important issue in the field, which is that of whether or not convergence in terms of gene expression would correlate with functional convergence at large evolutionary distances. This is a key question in the field that is often overlooked or assumed to be necessarily true. Our work is focused on the possible evolutionary relationships among cell types in the cephalopod and vertebrate visual systems rather than on “whether similar (convergent) processing” occurs in these two distantly related lineages. Although the question of convergent processing is a fascinating one, it is beyond the scope of our project. We now state this more clearly in the abstract “*Identification of an FMRF-amide-based retrograde signal from the optic lobe toward the retina, supports the ‘deep retina’ hypothesis from a molecular standpoint*” and introduction “*evaluate the ‘deep retina’ hypothesis (previously formulated from morphology) in terms of molecular signatures of cell identity*” to avoid confusion and clarify that we are not referring to visual signal processing.

In order to investigate cell types, we have examined how gene families have evolved and which genes from these gene families are expressed in which cell type. While we identify many homologous gene families between *E. berryi*, vertebrates and flies, we find that they are generally not co-expressed in the same cellular patterns across these three lineages. An exception is the co-expression of a battery of genes in both vertebrate retinal bipolar cells and a specific cell type in bobtail squid that we identify as JZ Young's morphologically-named ‘bipolar cell’ in cephalopods, we propose that this coexpression battery was convergently co-opted. We also investigate gene family expansions that took place in the cephalopod and spiralian lineages and observe independent expansions of genes that belong to the same gene families as those utilised by vertebrate neuronal tissues. We thus do demonstrate that independent gene family expansions have aided in the convergent building of the cephalopod visual system.

We would like to stress here that we distinguish between convergent evolution in terms of organ structure, function and gene expression, notions that Reviewer #1 is implying that we equate with each other. We do not expect convergent gene expression to necessarily correlate with convergent visual information processing mechanisms. This is laid out clearly in our manuscript: for instance, we observe a lineage-specific expansion of ionotropic cholinergic receptors in cephalopods while vertebrates have an independent lineage-specific expansion of ionotropic cholinergic acetylcholine receptors. The gene families in which each set of lineage-specific expansions took place are different. Moreover, the number of receptors found in cephalopods is considerably larger than those found in vertebrates. Although the gene families from which both these expansions originated were present in the common ancestor and the gene families are for the most part present in both lineages, the presence of these genes and their expression would not allow one to infer the visual signalling cascades employed by each species and how visual signal is perceived.

The suggestion that one molecularly defined cell type may correspond to Young's bipolar cells (which may not do anything like what the vertebrate bipolar cell does) or that the long-known centrifugal projection to the retina could use FMRFamide as a messenger, are interesting predictions, but are neither confirmed nor do they aid the argument for convergence.

There are many other conclusions, such as that their retina data confirm the relative simplicity of the cephalopod retina, but these are well-known, at least in the cephalopod field.

We identify a cell type in *E. berryi* (clusters Dopaminergic4 & 5) that, based on its gene expression and position in the medulla based on HCR staining, likely corresponds to the centrifugal cells that Young identified in his detailed morphological studies. We find that this cell type also synthesises FMRFamide. See also **Fig. 3c** for FMRFamide antibody staining in retina and **Fig. 3k, k'** for HCR staining of FMRFamide mRNA in optic lobe, demonstrating the FMRFamide peptide is synthesised in the optic lobe and is then transported to the retina. We argue that these constitute strong evidence for the molecular characterisation of centrifugal cells, and, to our knowledge, the first time that these points have been demonstrated. While previously cited work has described centrifugal cells and neuropeptide usage, they have not been linked together to our knowledge.

Regarding bipolar cells, cell types identified in two different groups of organisms (vertebrates and cephalopods) were independently named 'bipolar' based on cell morphology. We completely agree with Reviewer #1 that J.Z. Young's morphologically defined 'bipolar cells' in cephalopods do not *a priori* have any functional similarity in terms of visual signal processing to vertebrate 'bipolar cells', and our study does not tackle this. We have clarified this point in the text, as follows:

"These cholinergic cells are associated with Aces-2+ and Nkx21+ whose expression was localised to the medulla (Fig. 3g) and may correspond to the 'bipolar cells of the medulla' described by J. Z. Young. We note that this prediction is based on gene expression similarity and we do not imply that vertebrate and cephalopod bipolar cell types have a similar role in the visual signal processing circuitry."

and later:

"Future studies will be required to elucidate the position and role of these cell types in the cephalopod visual processing circuitry."

We agree with Reviewer #1 on the fact that the cephalopod retina has previously been described as being "relatively simple". Indeed in the introduction of our manuscript, we cite the electron microscopy and histology of J.Z. Young for the well-known relative simplicity of the cephalopod retina. We must however point out that the classical studies of the organisation of the cephalopod visual system have used octopus and loliginid squid. Here, we have introduced the genome-enabled *E. berryi* and performed single-cell transcriptomics and light and electron microscopy to explore the organisation of the bobtail squid visual system. As *E. berryi* is a new model whose visual system had not previously been investigated, we found it pertinent to first briefly show an overview of the system and introduce the retina. Effectively, as Reviewer #1

points out, the organisation of the *E. berryi* retina is not strikingly different from some of the other cephalopod models examined by other researchers.

The novelty of our results however is that we can identify the anticipated cell types of the *E. berryi* retina based on single-cell transcriptional profiles. Although the vertebrate retina is known to employ a relatively humble total of 7 cell types, recent studies based on single-cell gene expression have identified numerous cell sub-types in different vertebrate lineages. In contrast, we do not find a plethora of cellular subtypes in the bobtail retina. We do, however, find two photoreceptor cell type clusters in *E. berryi*. The morphological correlates of these two transcriptionally distinct clusters remain obscure, but it stands as a novel result.

Please note that we do not believe that the simple retina of *E. berryi*, the centrifugal projections and the putative bipolar cells are in themselves arguments for convergence, and have clarified this in the text. We start from the point of view that the visual systems of vertebrates and cephalopods evolved convergently, and our study considers the underlying genetic mechanisms through which this convergence took place.

This manuscript, available earlier this year on bioRxiv, joins other bioRxiv single-cell reports on cephalopods from the Sprecher, Niell and Seuntjens groups. These papers are cited but only just. In particular, the Seuntjens paper has a lot to say about octopus glial cells, and the Niell paper covers cell-type identity and development in the octopus OL. It might be profitable for the report under review to move beyond simple citation and highlight for the field what in their analysis aligns with the octopus findings, and what departs from it.

We thank Reviewer #1 for suggesting a comparison between the cell types recently identified in the different cephalopod lineages. We have now performed comparative analyses of our datasets with those reported in these studies. We now provide (i) two Supplementary Discussion notes (**Supplementary Notes 2 and 3**) on the reported marker genes across studies (ii) direct comparative analyses of these datasets using the widely-used cross-species integration SAMap methods (**Extended Data Figure 9**, see also below), (iii) contextualisation of our findings in the light of these comparisons in the Discussion of our revised manuscript (see below) and (iv) an interactive web-based Shiny app (http://141.164.60.190:3838/Eberryi_visual/) that makes it easier to explore and compare the different single-cell datasets.

It is however important to note that these datasets are not all directly comparable to ours. Here, we comprehensively profile single-cell transcriptomes in specific organs (15,000 to 20,000 cells for each of our retina, adult optic lobe and hatchling optic lobe datasets). In the other studies:

- Songco-Casey et al. profile 28,855 cell transcriptomes in the optic lobe of a juvenile octopus *O. bimaculoides*. This dataset is directly comparable to our optic lobe data, and is relevant to identify cephalopod-conserved cell types since *E. berryi* and *O. bimaculoides* descend from the two most deeply-branching cephalopod clades (Octopodiformes and Decapodiformes, divergence ~300-400Mya).
- *Styfhals et al.* profile 17,081 cell transcriptomes from the hatchling octopus *Octopus vulgaris* whole brain (comprising vertical, subesophageal and optic lobes). This dataset is

somewhat comparable to our optic lobe datasets but does not comprehensively capture the cell populations in this organ.

- *Duruz et al.* profile 20,000 cell transcriptomes in the whole head of a pre-hatchling squid *Loligo vulgaris*, incl. brain and eyes. This dataset is somewhat comparable to our retina and optic lobe data, but from a different developmental stage and with the caveat that the data do not comprehensively capture the cell type diversity in these organs.

Conclusions of these new comparisons can be found in the Discussion:

To begin to draw general rules regarding the molecular underpinnings of cephalopods visual systems, we conduct comparison with recently published single-cell datasets in other cephalopods (Songco-Casey et al., Styfalhs et al., Duruz et al.) (Extended Data Fig. 9, Supplementary Discussion Notes 2-3). Notably, comparison with optic lobe octopus single-cell transcriptomes (Songco-Casey et al. 2023) demonstrate the broad implications of our findings (Extended Data Fig. 9). First, diversification of dopaminergic neuronal cell types is a global signature of cephalopod optic lobes. Second, the FMRF+ centrifugal cell population that we identify is molecularly conserved across cephalopods (Dopaminergic5-6 matching Ob.dopa.fmr.10-11), as are the putative cephalopod bipolar cells (Cholinergic1 and Glutamatergic1 matching Ob.chol.med.otx.8 and Ob.glut.med.30-31) and dopaminergic neurons molecularly related to vertebrates ciliary photoreceptors (Dopaminergic 1-3 matching Ob.dat.glut.IGL 5-6). Additional analyses including octopus hatchling brain single-cell transcriptome (Styfalhs et al., 2023) uncover strong conservation of glial gene expression (Extended Data Fig. 9a-b) and highlight conserved markers of cephalopods glia (Eaa1-2 and Glna-2). As cephalopods do not express transcription factors characteristic of arthropod or vertebrate glia, we however suggest independent evolution of glial cells in these lineages (Hartline 2011). Finally, comparisons with loligo squid whole head from Duruz et al., 2023 highlight conservation of cephalopods broadly defined photoreceptors and lentigenic cells.

Extended Data Figure 9. Comparisons of *E. berryi* visual system single-cell transcriptomes with single-cell transcriptomes from other cephalopods. a-d, Heatmaps showing SAMap single-cell expression comparisons between cell populations of *E. berryi* adult optic lobe and **a, *O. bimaculoides* optic lobe, **b**, *O. vulgaris* hatchling whole brain, **c**, *L. vulgaris* pre-hatchling whole head, **d**, between *E. berryi* adult retina and pre-hatchling *L. vulgaris* whole head.**

Rebuttal Figure: Screen capture of the interactive web-based shiny app to explore single-cell datasets.

Reviewer #2 (Remarks to the Author):

Gavriouchkina and co-authors introduce a single-cell transcriptomic atlas of the bobtail squid visual and nervous systems. They combine this extensive single-cell sampling with a high-quality genome assembly and annotation (an outstanding resource), as well as extensive imaging and HCR validations. In addition to the high quality of the data, I really appreciate the way the results are carefully contextualized and interpreted.

Besides the interest that this study has for understanding cephalopod biology, the studied system represents an interesting case-example of phenotypic convergence and the authors take this opportunity to explore the molecular signatures underlying this process. Unfortunately, in my opinion this comparative aspect of the manuscript is lacking in details and would benefit from a more thorough and quantitative analysis. Overall, though, I think this an excellent and interesting study and a valuable resource. Therefore, I find it suitable for publication in NEE.

We sincerely thank Reviewer #2 for the positive and helpful comments!

These are my main criticisms/comments, which I hope can help improve the clarity and consistency of the paper:

We would like to thank Reviewer #2 for these suggestions to improve clarity of the single-cell RNA-seq analyses.

- Have the authors tried to jointly cluster all datasets? It should be a more elegant way of presenting the results (e.g. sorting out blood cells together from the retina and control blood experiments) and then zooming into interesting subpopulations as needed – as it is commonly done in single-cell studies.

We have indeed tried to cluster all datasets together, but opted to retain separate representations and analyses in the main text, since the organs we study present a rich variety of cell (sub)types and the majority of the cell-types we report are organ-specific. We found that integration of these datasets does not work very well and projecting them onto a joint 2D space results in a UMAP that is complex to read and poorly informative.

See for instance the integration of 2 of our 4 datasets together: adult optic lobes and Retina (Rebuttal Figure below). The blood (and white body) cell populations do indeed cluster together across datasets, as expected, but other cell clusters are organ-specific.

We have not attempted to co-clustered the blood from each of the samples, as hemocytes were not the focus of our study. As we are not experts in cephalopod hematopoiesis, we provide this data for the qualified community to explore. To aid reviewers and readers to better navigate through our dataset, we offer a Shiny app allowing one to visualise individual gene expression and review the differentially expressed markers of each cluster:

http://141.164.60.190:3838/Eberryi_visual/

Rebuttal Figure: UMAP projection of adult optic lobe and retina single-cell expression datasets after integration. Cell population mixing across the two organs can be observed on the leftmost UMAP: very few clusters of each dataset are mixed (with the exception of blood, on the right). The two rightmost UMAP show cell clusters identified in the adult optic lobe and retina, respectively.

Finally, we concur that analysis of the hatchling and adult OL datasets together is important and have now included it (see more details in our response to the specific comment on optic lobes below).

- Related to the previous point: how batchy is the single-cell data from different preparations and 10x runs?

We thank the reviewer for raising this important point. We had indeed successfully integrated single-cell data from biological replicates (same organ and developmental stage). We show below the UMAP projections of replicates for each integrated dataset, which we have now also included in Figure S2. Importantly, the cell clusters that we report are consistently found across replicates, and cell proportions are overall similar (Figure S2).

RETINA

ADULT OPTIC LOBE

HATCHLING OPTIC LOBE

BRAIN

Selected panels of Extended Data Fig. 2: UMAP projection of single-cell expression datasets after integration across biological replicates, for retina, adult optic lobes, hatchling optic lobes and brains (top to bottom).

- I wonder if there are any genes exclusively expressed in the “photoreceptor_r2” population. In contrast to this, and looking at Figure S3, I find it very strange that the tiny “photoreceptor_undiff” population expresses so very many specific markers, how do the authors explain an undifferentiated/progenitor population with so many and so specific effector genes?

We agree, and have now renamed “photoreceptor_undiff” as “photoreceptor_r2b.” As reviewer #2 point out, it expresses specific markers of differentiated photoreceptors. It does, however, also express musashi, which led us to think that it might represent a stem-cell-like or progenitor population. This cell population is small, and we don’t have HCR markers specific for it. (We tried

a probe designed against the Musashi gene but it didn't show any interpretable signal.) So as suggested by the reviewer, we decided to limit our claims for a specific role for these cells in our study.

Regarding the photoreceptor_r2 population, it is correct that few of the markers we report (**Fig. 2b and Table S4**) are specific to this cell-type, namely: Kcna1 (regulation of neuronal excitability), Cd5r1 (neuronal development) and Vigilin (cholesterol detoxification). We clarify in the text that functional interpretation of the 2 photoreceptor cell clusters remain elusive. The most striking difference relates to the fact that only photoreceptor_r1 express both r-opsins 1 and 2, retinol-binding protein, dopamine receptor and glutamate transporter, whereas photoreceptor_r2 expresses only r-opsin2.

- Continuing with the retina dataset, any idea what is the “contractile” population? What makes the authors think this is contractile?

We had previously called these cells “contractile” because they show low-level expression of orthologs of sarcomeric myosin and troponin. Upon further reflection, it is more likely that these cells are comparable to fibroblast cells, as suggested by the strong expression of an alpha collagen. We have renamed the cell cluster accordingly.

- The analysis of adult vs 1dph hatchling datasets (Figure 4) is rather limited. The authors should either analyze them together or show a pairwise comparison (e.g. using gene expression correlations or jaccard intersections of shared markers) between these two datasets.

We thank the reviewer for this suggestion and agree that inclusion of more direct comparisons of these datasets is important. Since these two datasets contain a large number of (distinct) cell (sub)types, we found separate clustering and analysis to better capture their complexity. We however now also present these datasets projected into a joint UMAP space after integration and assess expression similarity using correlation (Figure S6, see below). This new analysis further supports our conclusion of distinct neurotransmitter usage, even across “putatively matched” cell clusters of the two optic lobe datasets. While neurotransmitter usage switch is clearly demonstrated by our data, we remain cautious over the interpretation of cell cluster matches across hatchling and adult optic lobe datasets: more scRNA-seq time points throughout optic lobe maturation combined with experimental lineage tracing would more precisely capture cell type dynamics as well as the precise timing of neurotransmitter usage switch.

Selected new panels from Extended Data Fig. 6: UMAP projection of integrated **a**, adult and **b**, hatchling optic lobe single-cell data. Integration was performed using the standard Seurat SCT integration workflow. Note that clustering was performed separately, on non-integrated data (see main Fig. 3 and 4). Coloured cluster labels highlight cell clusters with distinct neurotransmitter usage in adults as opposed to hatchlings. **c**, Gene expression similarity across clusters of the adult and hatchling optic lobe datasets, using weighted Pearson correlation. Cell cluster colours at the left (adult) and top (hatchling) of the heatmap match the colours used in the UMAPs.

- The cross-species comparisons would benefit from a less descriptive/more quantitative approach. For example, identifying how many and which specific gene orthologs (or paralogs of the same homology group) are shared between clustered chicken-squid cell types, including TFs. We never get to see the co-expression of markers described by the authors. One example of how the paper would benefit from this is the “bipolar-cell gene expression module” that the authors describe in the discussion – can you retrieve the genes that would compose this module from your cross-species analysis?

As suggested by the reviewer, we now include a table (**Supplementary Table 10**) reporting how many and which specific genes are shared between clustered chicken-squid cell types, including TFs.

Regarding the “bipolar-cell gene expression module”, we have clarified the text to better reflect our observations. We do not observe global gene expression conservation between vertebrate bipolar cells and cell type of the cephalopod optic lobe. However, focusing on traditional, experimentally-validated markers of vertebrate bipolar cells, we observe that their expression is specifically restricted to putative cephalopod bipolar cells of the optic lobe medulla (glutamatergic1 and cholinergic1 cell populations). This conserved gene expression module is presented in **Fig. 5b**. We propose that the absence of global expression similarity indicates a convergent recruitment of these genes that are important for bipolar neuron function and thus the convergent evolution of bipolar neurons in vertebrates and cephalopods.

- Similarly, the paper is missing a quantitative perspective on the “subfunctionalization” question, which remains highly speculative/descriptive. I would recommend omitting this bit or providing a dedicated quantitative analysis and visualization to support this (otherwise interesting) point.

We thank the reviewer for the suggestion. It is correct that our previous analysis only demonstrated an enrichment in cephalopod gene duplicates being expressed in specific cell-types (**Fig. 5g,h**), not that duplicated copies of the same families contributed to distinct cell populations. To investigate this question, we introduced a new analysis where we count the number of gene families in which distinct paralogues are expressed between pairs of clusters (**Extended Data Fig. 8c** and below). While the majority of paralogs are expressed in the same cell type (heatmap diagonal), this result shows the contribution of distinct paralogs between some clusters, predominantly differentiating clusters (Neuro1 and 2) but also some of the dopaminergic cell clusters (Dopaminergic1, 2 and 3). We have included this new analysis in the manuscript and rephrased to clarify that we cannot know whether novel cell-type specificity of paralogs evolved through neo- or sub-functionalisation.

Extended Data Figure 8c. Contribution of cephalopod paralogs to *E. berryi* optic lobe cell type diversification. Numbers in the heatmap indicate the number of cephalopod paralogous gene families for which distinct paralogs are specifically expressed in distinct cell populations.

- I would like to see more details in the methods, particularly regarding gene orthology and cross-species comparison (e.g. the motivation behind using Metaneighbors and the consistency of alternative approaches).

We have revised the Methods to include more details regarding the MetaNeighbor approach and gene orthology (see Method section “Cross-species comparisons of single-cell data”). Using an alternative approach - the more recently proposed SAMap method based on single-cell dataset

integration - we now confirm our main results that *E. berryi* optic lobe cell populations Dopaminergic1 and 3 consistently display similarity with vertebrate cell-types across the different comparisons and methodologies (new **Extended Data Fig. 7e**). We retain our MetaNeighbor approach as the analysis presented in the main text, because it allows us to have more control on the orthologous genes leveraged in comparisons (and presents expression-based cell type clustering across the two species). In contrast, SAMap leverages all many-to-many homologs as identified through Blast hits, potentially also including distant out-paralogs and giving disproportionately elevated weights to highly duplicated genes during manifold stitching (for instance, the highly duplicated cephalopod protocadherin genes or GPCRs).

- The authors MUST unify the way gene expression (both single-marker plots and multi-marker maps) across cell types is visualized, e.g. cluster-level normalized expression in Figure S2e-f; % expressing cells dotplots in Figure S3; violin plots in Figure 3; and single-cell-level normalized expression in Figure 1. Figure S4 is particularly chaotic example of this. Choose one style and stick to it.

We thank the reviewer for the feedback. We have unified visualisations of single-cell gene expression across all main and supplemental figures: we only use cluster-level normalised expression. We use dotplots as the main representation, with the only exception of 2 violin plots in Figs. 2c and 3b (these are the key genes and markers that help us annotate cell identity and we want to highlight them). Gene expression from bulk RNA-seq in Supplements is visualised in heatmaps, to differentiate them from scRNA-seq.

- Please, make sure all figure legends and axes are labeled (e.g. Figure S2, y-axis in Figure 3b). We have carefully checked the figure and added labels for axes when missing.

- Small typo: Line 939: “protomes” -> “proteomes”

Thanks for catching this typo.

Reviewer #3 (Remarks to the Author):

This paper introduces the genome and single cell transcriptomic analysis of the cephalopod, *Euprymna berryi*. The authors carried out single cell sequencing for cells of the retina, optic lobe and non-neural tissues for adult and juvenile animals. Neuronal and other cells were identified by canonical markers, and neurons were subdivided based on transmitter expression/reception. The work establishes a rich and valuable resource for developmental-genetic studies of cephalopod neural development. The authors also use the data to confirm previous findings, in particular the cell typing of the retina, the idea of considering the optic lobe as a “deep retina”, and peptidergic feed back innervation of the retina by the optic lobe. The paper adds significant new insights into the development and evolution of the bilaterian visual system. Findings are generally well described and documented.

We would like to thank Reviewer #3 for the positive comments and detailed suggestions.

My comments:

1. In the introduction and throughout the Results sections the authors need to provide more context when presenting details of cephalopod neuroanatomy.

We thank Reviewer #3 for this suggestion. We have now added a new figure panel (**Extended Data Figure 4a**) that summarises HCR results in the context of the neuroanatomy of the cephalopod optic lobe (see also our reply to the specific comment 1.3 below). In our present study, we included microCT images of the optic lobe (**Figure 1c**) that are consistent with prior observations in cephalopods, demonstrating that *E. berryi* follows the classical cephalopod organisation.

1.1. Introduce apparent simplicity in terms of photoreceptor types; given the large cephalopod retina (millions of photoreceptors), why are there only 1-2 receptor types? How is color vision achieved? Have previous works commented on this paucity of photoreceptors (as well as support cells)?

These are indeed quite fascinating questions! How cephalopods perceive colour has been a matter of much debate and theory over the years. Cephalopods are well known for their ability to change colour to camouflage themselves, suggesting that they can both perceive and thereafter reproduce observed colour. However traditionally most studies have not identified more than one visual pigment or more than one opsin gene (Brown and Brown, 1958; Hara and Hara, 1972; Muntz & Johnson, 1978), with the exception of firefly squid *Watsenia scintillans* is known to host two visual pigments with a max absorbance at 470nm and one at 500nm (Matsui *et al.* 1988, Seidou *et al.*, 1990, Michinomae *et al.*, 1994). More recently, Bonadè *et al.* cloned two distinct r-opsin genes in *Sepia officinalis* however only one of these opsin genes was expressed in the retina (Bonadè *et al.*, 2020). The cephalopod retina has therefore traditionally been thought to be “colour blind”.

To explain cephalopods' apparent ability to discriminate colour without distinct visual pigments and opsins, several hypotheses have been put forward. Some authors have evoked the possibility of filters permitting light discrimination, similar to those of the butterfly *Heliconius erato* which employs photoreceptors with only one opsin but carries out pigment colour discrimination using different peri-rhabdomal filter pigments (Zaccardi et al. 2006). Body skin chromatophore and iridiophore expansion and retraction has been proposed to function much in a manner reminiscent of oil droplets function in colour vision of animals such as turtles (Liebman & Granda, 1975). The theory that has gained the most support is that cephalopods are able to discriminate polarised light through both their retinas and through specialised cells on the skin (Moodey & Parriss, 1960, 1961; Rowell & Wells, 1961, Shashar & Cronin, 1996, Tasaki & Karita, 1966, Saidel et al., 1983, Shashar et al., 2001, Mäthger *et al.*, 2009). Yet another hypothesis suggested is that the unusual shape of some cephalopod pupils could allow them to discriminate colours based on chromatic aberration (Stubbs and Stubbs, 2016).

In *E. berryi*, however, we find two opsins that are expressed in two molecularly distinct cell types that may be morphologically identical or very similar. Thus, it is possible that the *E. berryi* retina may be able to distinguish colours. To our knowledge, one other study has identified two opsin molecules (and two retinochrome molecules) in a cephalopod genome, in the cuttlefish *Sepia officinalis* (Bonadè et al. 2020). However when Bonadè et al. used traditional *in situ* hybridisation (which is less sensitive than HCR or single-cell sequencing methods), they only observed the expression of one of the opsins and both retinochromes in the cuttlefish retinas. In *E. berryi*, we observe the expression of one retinochrome and two opsins (see Fig. 2). To our knowledge, there are no other reports of two opsins being expressed in a cephalopod retina. It is possible that the expression of two opsins is a peculiarity of *E. berryi* and that other cephalopods do not use both their r-opsins in their photoreceptor cells.

We had not originally insisted on these exciting aspects of visual signal colour discrimination in our manuscript because cephalopods also have extra-ocular opsin expression in their skin (Kingston 2013, 2015, 2016; Young 1979; Young & Mencher 1980) hypothetically 'to measure downwelling light and adjust ventral counterillumination' (Mäthger et al., 2010). We cannot however at present pronounce ourselves on whether or not *E. berryi* employ these two opsin genes to perceive more than one colour. We now mention this in the discussion.

1.2. The support cells appear to produce pigment. What about glial cells in the retina?

We thank Reviewer #3 for this question, which we have in part addressed in our answers to questions from Reviewers #1 and #2. The very name of the 'support' cells may lead one to believe that support cells of the retina may be glial in nature. However, very little is known about the enigmatic support cells of the cephalopod retina. There are no known markers of cephalopod retinal support cells described in the literature to our knowledge (Yamamoto et al., 1965; Young et al., 1971; Lange & Hartline, 1974; Yamamoto et al., 1984; Koenig et al., 2015). Support cells have been described as cells lacking rhabdoms intermingled with photoreceptor cells. Electron microscopy studies by Yamamoto et al. (1965, 1984) noted a 'heavily pigmented supporting cell', while J.Z. Young (1971) described one type of supportive glial cell containing pigment and one

type of glial cell in the nuclear layer exterior the the basement membrane. Our EM evaluation of the *E. berryi* retina did not reveal any additional cells in the retina proper (see Fig. 2). We proceeded by elimination to identify the support cell population. The Pigmented2 population expressed markers of pigmentation such as Aspartate aminotransferase Aatm ('white') gene and similar levels to tryptophan 2,3-dioxygenase T23o as in the photoreceptor cells and did not express microvillus markers crumbs, Crb. Curiously, the Pigmented2 cells appeared to express r-opsin2 and many members of the phototransduction cascade (see Fig. 2), but did not express retinochrome-2 or visual cycle genes. We therefore concluded that the identity of Pigmented2/support cells were closer to that of photoreceptors than to glial cells. Pigmented2/support cells did not express the glial marker Eaa1-2 that we, Duruz *et al.*, Styfhals *et al.* and Songco-Casey *et al.* have accepted as a bone fide glial marker in cephalopods. Pigmented2/support cells did however express glutamine synthetase enzyme Glna-2, which was also expressed by other cells in the retina (including photoreceptor cells, lentigenic cells, endothelial and pigmented Tyro1+ cells). Yamamoto *et al.* noted that support cells in the *Octopus* retina carried desmosome-like or gap junctions in their EM study. We did not observe markers of gap junctions in the pigmented2 population markers.

Whether these cells constitute 'glial' is a difficult question to answer as it depends on what one means by glial. As described in **Supplementary Discussion 2**, glia have evolved several times in multiple lineages, expressing many different markers (for example in arthropod lineages as opposed to vertebrates). Our co-clustering analysis detailed in our response to comments from Reviewers #2 shows that glial cells identified in the optic lobe do not co-cluster with the pigmented2 cells of the retina.

1.3. Summarize existing structurally/transmitter defined neuron types of the optic lobe.

A schematic showing the organization of the optic lobe, with layering (cell bodies vs neuropil) and structurally defined cell types, similar to the schematic of the retina in Fig.2h, would be very important. How is the medulla organized? It does not seem layered (Fig.3); cell body clusters appear to be interspersed with domains of neuropil.

The classic work of JZ Young showed that while the cortex of the cephalopod optic lobe may have a layered structure (confirmed by our data), the medulla is organised differently, with clustered cell bodies. We have now added a new figure panel (**Extended Data Figure 4a**, see below) that presents neuronal organisation in the optic lobe. In this Figure, we also indicate the predicted localisation (as inferred from HCR experiments) of the cell populations that we have molecularly characterised.

Selected new panel from Extended Data Figure 4. a. Schematic optic lobe neuroanatomy and summary of cell localisation from HCR (see Fig. 3). Abbreviations: OPL - outer plexiform layer, OGL - outer granule layer, IPL - inner plexiform layer, IGL - inner granule layer, PL - palisade layer, CE - centrifugal.

1.4. Mainly transmitters are used to differentiate neuron classes. Has the expression/localization of these transmitters been studied based on (immune)histochemistry in cephalopods in previous studies? Have these transmitters been assigned to morphological cell classes? If so, do the current localization findings confirm or contradict existing studies?

Regarding the correspondence between neurotransmitters and cell types, the assignment of neurotransmitters to specific morphological classes has not been done. Most of the knowledge we have on cephalopod visual system neuroanatomy comes from the detailed morphological studies of JZ Young. JZ Young did not conduct any immunohistochemical studies that would allow linking cell morphology and localization together with neurotransmitter usage. We have added to the revision a referenced **Supplementary Discussion Note S1** text summarising the literature on neurotransmitter usage in cephalopods and how it aligns with our findings.

However, our work and recent single cell studies in other cephalopods represent a first step in this direction. To provide a broader picture of results from these different studies regarding cell molecular identity, cellular localisation and neurotransmitter usage, we now contextualise our results based on findings in these recently published studies in the Discussion section of our revised manuscript (see also **Extended Data Fig. 9**). We have also developed a web application (http://141.164.60.190:3838/Eberryi_visual/) to make it easier to explore molecular markers for each cell type.

1.5. Providing more background on neuroanatomy is particularly important in regard to interconnections between photoreceptors and optic lobe neurons. In the vertebrate (or arthropods) there is a strict layering, where photoreceptors project on to bipolar cells, which in turn target the output neurons. The authors cite previous neuroanatomical findings that “bipolar cells” were identified in cephalopod optic lobes. These findings need to be summarized/presented schematically (see above). Does the localization of putative molecular markers for bipolar cells match the descriptions by Young?

JZ Young designated some medullary cells of the cephalopod optic lobe as “bipolar” based on their cellular morphology (i.e., possessing two axonal projections from opposite sides of the cell body). Unfortunately, as discussed in our response to the previous comment, this early work was all about cellular morphology and connectivity, and JZ Young did not do any immunohistochemical studies that would provide additional information. However, based on our HCR stainings, the cells that we putatively identify as bipolar in our single-cell transcriptomes are consistent with bipolar cell location reported in JZ Young’s drawings.

2. Glia: Glial cell types have been characterized in many bilaterians by morphology and interaction with other cell types; examples are the astrocytes/astrocyte-like cells in vertebrates and insects (interacting with neurites and synapses and synapses in the neuropil), oligodendroglia (myelin sheaths) in vertebrates, cortex glia (ensheathing cell bodies) in insects, etc. Please provide an overview/schematic of existing studies (EM analysis and others) that characterize glia in cephalopods

We thank the reviewer for raising this. Similarly as *Drosophila*, cephalopods do not have myelin sheath, but they do have (some) glia ensheathing cell bodies. We have now provided a detailed summary of previous histological studies of glia in cephalopods in **Supplementary Discussion Note 3**. Briefly, these studies have identified three types of glia in cephalopods: (1) glial cell associated with blood vessels, (2) protoplasmic glia and (3) fibrous glia. In our HCR stainings, we observe expression of Eaa1-2 glial markers in both large cells situated in the palisade layer of the medulla (see Fig. 3j-j’, arrowheads) and in a punctate pattern in the inner plexiform layer. We propose that the larger cells may correspond to fibrous glia, whereas the punctate pattern may correspond to protoplasmic glia. We mention this in the main text and refer to the new supplementary note for details.

3. Evolution:

3.1. The authors place their findings in the framework of the “deep retina” hypothesis, whereby vertebrate retinal neuronal layers are “displaced” into the optic lobe. This idea has been discussed many times also in regard to arthropod eyes in which, like in cephalopod eyes, the retina is formed only by photoreceptors, and no interneurons. An inference from the deep retina hypothesis would be that during vertebrate evolution, brain neurons (e.g., interneurons

targeted by photoreceptors) become “outsourced” into the retina. An alternative view has been presented (e.g., Arendt: Evolution of eyes and photoreceptor cell types. Int J Dev Biol 2003) that postulates that at the onset of eye evolution, photoreceptors carried out all visual function; receptors then diversified into different types, as well as neurons. Gene expression data were also used to defend this “shallow retina” hypothesis. Given the scope and importance of the current paper for the cephalopod visual system and visual system evolution in general, this alternate hypothesis needs to be exposed in the introduction. Are there data that would more clearly help (or defeat) the shallow retina hypothesis?

We thank the reviewer for sharing this interesting perspective and for commending the importance of our paper for the cephalopod visual system evolution field. We concur on the relevance of placing our work in the context of the propositions formulated by Arendt on the evolution of vertebrate retinal cell types.

We now expose the model proposed by Arendt and discuss our findings within that framework (see more specific reply in response to comment 3.2 below). We however refrain from naming these two models deep and shallow retina hypotheses: these models relate to an evolutionary scenario for eye evolution and are distinct from the “deep retina” proposition of JZ Young that only pertained to organisational similarities (feedback from cephalopod optic lobe neurons to the photoreceptors in the retina, similar to feedback from vertebrate retinal neurons to photoreceptors).

3.2. For example, in line 448, the authors mention a similarity in gene expression between vertebrate retinal ganglion cells and squid photoreceptors: does this not support the shallow retina hypothesis (photoreceptors diversifying into neurons that form deep layers of retina)?

The reviewer is correct. We have now integrated this observation and have deeply thought our interpretations to discuss support brought by our data for the model proposed by Arendt, see the following paragraphs of our revised manuscript:

Two models have been previously proposed to explain cell type evolution at the origins of complex visual systems, both based on quantification of a few proposed key marker genes (Erclik et al. 2009). In a first model (Arendt et al. 2016), all vertebrate retinal cells are proposed to have arisen from two spatially and molecularly distinct ancestral photoreceptors, forming two groups: cells related to ciliary-Photoreceptors (Bipolar cells) and cells related to rhabdomeric-Photoreceptors (Retinal Ganglion cells, Horizontal cells and Amacrine cells). In a second model (Erclik et al. 2008), the ancestral eye already contained several types of photoreceptors, their target Vsx+ interneurons and projection neurons. In the vertebrate lineage these three primitive cells evolved to become ciliary photoreceptors, bipolar cells and ganglion cells, respectively, whereas in the fly lineage, rhabdomeric photoreceptors remained in the eye and the ancestral interneuron cell type and projection neuron expanded to the Drosophila optic lobe. Note that both models can be compatible: combined together they predict that the interneuron and

projection neurons of the ancestral eye have derived from ciliary and rhabdomeric photoreceptors, respectively.

Our expression comparisons between chicken retinal cell types and the *E. berryi* optic lobe (**Fig. 4a**) and retinal cell-types (**Extended Data Fig. 7a**) recover two main clusters, one cluster of cell types associated with ciliary photoreceptor and a second with rhabdomeric photoreceptors, thus supporting model 1 from a genome-wide standpoint. Interestingly, while the majority of the *E. berryi* optic lobe neuronal subtypes are more similar to rhabdomeric photoreceptors, we also find a dopaminergic cell population (Dopaminergic1,3) with strong expression similarity to vertebrate ciliary photoreceptors and bipolar cells (**Fig 4a, Extended Data Fig. 7e**). This observation suggests that the cell types of the cephalopod visual system, similarly as vertebrates, may have been assembled by building upon both ciliary and rhabdomeric photoreceptors. Moreover, in line with the proposed conservation of a conserved *Vsx+* interneuron module, we also observed specific expression of orthologues of general vertebrate bipolar cell markers (*Vsx2*, *Otx2*, *Lhx3*) in the bobtail squid optic lobe clusters *Glutamatergic1* and *Cholinergic1* (**Fig. 5b**). Notably, Cone Bipolar OFF-centre cell type marker orthologues were enriched in *Cholinergic1* of *E. berryi*. These cholinergic cells are associated with *Aces-2+* and *Nkx21+* whose expression was localised to the medulla (**Fig. 3g**) and may correspond to the ‘bipolar cells of the medulla’ described by J. Z. Young. We note that this prediction is based on gene expression similarity and we do not imply that vertebrate and cephalopod bipolar cell types have a similar role in the visual signal processing circuitry.

Surprisingly, conservation of these selected markers did not translate into global expression conservation between bobtail squid *Cholinergic1*/*Glutamatergic1* and vertebrate bipolar cells. *Cholinergic1*/*Glutamatergic1* globally resemble more vertebrate amacrine cells (*Cholinergic1* matching a subtype of GABAergic amacrine cells and *Glutamatergic1* matching a subtype of Glycinergic amacrine cells) and as noted above, it is *Dopaminergic1-3* that resemble vertebrate bipolar cells and ciliary photoreceptors (**Fig. 4a and Extended Data Fig. 7e**). We therefore speculate that conservation of a *Vsx* module might be better explained by convergent reuse of these genes that are likely important for bipolar cell function, rather than homology. In light of these observations, we propose the following scenario for the evolution of the cephalopod visual system, consistent with both models presented above: two types of photoreceptors were present in the ancestor, along with two neuronal populations: one derived from ciliary photoreceptor and giving rise in cephalopods to *Dopaminergic1-3* cells, and one derived from rhabdomeric photoreceptors and giving rise to cephalopods *Cholinergic1* and *Glutamatergic1*. Future studies will be required to elucidate the position of these cell types in the cephalopod visual processing circuitry.